



**Validation of SNPP OMPS limb profiler version 2.6 ozone profile**
**retrievals against correlative satellite and ground based**
**measurements**
Nigel A. D. Richards[1,2], Natalya A. Kramarova[2], Stacey M. Frith[3,2], Sean M. Davis[4] and Yue Jia[5]
[1]Goddard Earth Sciences Technology and Research (GESTAR II), University of Maryland Baltimore County
(UMBC), Baltimore, MD, USA
[2]NASA Goddard Space Flight Center, Greenbelt, MD, USA
[3]Science Systems and Applications, Inc., Lanham, MD, USA
[4]NOAA Chemical Sciences Laboratory, Boulder, CO, USA
[5]University of Texas at Dallas, Dallas, TX, USA
*Correspondence to:* nigel.richards@nasa.gov
**Abstract.** The Ozone Mapping and Profiler Suite Limb Profiler (OMPS LP) was launched onboard the Suomi National
Polar-orbiting Partnership (SNPP) satellite in 2011 and began routine science operations in April 2012. The OMPS
LP uses measurements of scattered solar radiation in the ultraviolet, visible and near infrared wavelengths to retrieve
high vertical resolution profiles of ozone from 12 km (or cloud tops) up to 57 km. In mid-2023, version 2.6 of the
OMPS LP ozone profile retrievals was released, featuring improvements in calibration, the retrieval algorithm, and
data quality. We evaluate OMPS LP version 2.6 ozone retrievals using correlative data from other satellite instruments
and ground based data for the period April 2012 to April 2024. Our results show agreement between OMPS LP and
all correlative data sources between 20 and 50 km at all latitudes with differences of less than 10%, with OMPS
generally exhibiting a negative bias, except between 32 and 38 km in the tropics and southern mid-latitudes, where
the bias is positive. In the tropics and southern mid-latitudes the differences between OMPS LP and MLS, and OMPS
LP and SAGE III/ISS are less than ±5% between 20 and 45 km. Above 50 km, the agreement with MLS is still on the
order of -5% or better. Larger positive biases, up to ~35%, are seen in the upper troposphere lower stratosphere layer
(~15 to 20 km) between approximately 40° South and 40° North. We find that OMPS version 2.6 ozone exhibits the
same seasonal cycle as compared to all correlative measurement sources and our analysis shows that there is no
significant seasonal bias in the OMPS LP. We find small drifts relative to correlative observations at all latitude bands
of less than ±0.2%/yr (±0.1%/yr) between 25 and 50 km for the 2012-2024 period, with larger drifts above 50 km and
below 20 km. These small drifts vary between correlative measurements and straddle the zero line, we therefore
conclude that there is no significant systematic drift in OMPS LP version 2.6 ozone for the period 2012 to 2024. The
drift results represent an improvement in the long term stability of version 2.6 ozone over that of version 2.5.
**1. Introduction**
Stratospheric ozone is crucial for life on Earth as it acts as a protective layer absorbing harmful UV radiation. In 1985,
the discovery of the Antarctic ozone hole (Farman et al., 1985) caused global public safety concerns, ultimately leading
to the establishment of the Montreal Protocol in 1987. The regulations imposed by the Montreal Protocol have led to
a slow recovery in upper stratospheric ozone over the 2000-2020 period. Measurements show a positive trend in upper
stratospheric ozone in the range of 1.5-2.2% decade[-1] outside of the polar regions at mid-latitudes in both hemispheres
and 1.1-1.6% decade[-1] in the tropics (WMO, 2022). These increases are consistent with model simulations that show
they arise from a combination of decreasing ozone-depleting substance concentrations and decreases in stratospheric
temperature driven by increases in $CO_2$ (WMO, 2022). Conversely, there is evidence from both observations and
models for a small decrease in tropical lower stratospheric ozone over the same time period of 1-2% decade[-1.] This
decrease has a large uncertainty of ±5% decade[-1], but is consistent with climate change-driven acceleration of the
large-scale circulation and has a small impact on total column ozone (WMO, 2022). Observations and models disagree
on the sign of the trend in lower stratospheric mid-latitude ozone as ozone in this region has large year-to-year
variability and so trends have large uncertainties (WMO, 2022).



In order to detect such ozone changes, and to continue to monitor the health of the ozone layer, long term, vertically resolved, global observations of stratospheric ozone are needed. The NOAA/NASA Ozone Mapping and Profiler Suite (OMPS) sensors are a series of satellite instruments that are designed to meet this need by providing both total ozone and profile measurements (Flynn et al., 2006). The OMPS consists of three different sensors: a nadir mapper (OMPS NM), a nadir profiler (OMPS NP) and a limb profiler (OMPS LP). The first OMPS was launched onboard the Suomi National Polar-orbiting Partnership (SNPP) satellite in 2011 and consisted of all three OMPS sensors (Kramarova et al., 2014). The second was launched onboard NOAA-20 in 2017 with just the NM and NP on board, and the third, which again consisted of all three sensors, was launched onboard NOAA-21 in 2022. Two more OMPS containing all 3 sensors will be launched in the next decade providing decades of continuous ozone observations.

The validation of remotely sensed observations is crucial, not only to give confidence in scientific conclusions drawn from their use, but to also build community trust in the data and thus encourage their wider use. For this reason, when validating such data, we need to compare the retrieved data to as many different sources of correlative observations as are available to us. In this study we validate OMPS LP version 2.6 ozone retrievals against ozone profile data from two solar occultation satellite instruments (SAGE III/ISS and ACE-FTS), limb emission satellite Aura MLS, the nadir viewing satellite OMPS NP, a set of ground-based ozonesondes, and the lidar at Mauna-Loa.

## 2.   The OMPS Limb Profiler

The Ozone Mapping and Profiler Suite Limb Profiler (OMPS LP) is a series of satellite sensors that perform limb measurements of scattered solar radiation in the ultraviolet, visible and near infrared wavelengths (290 to 1000 nm) (Kramarova et al. 2014) which allow for the retrieval of ozone profiles from the top of clouds up to 57 km. In order to increase the cross-track coverage, the OMPS LP instrument has three observation slits separated horizontally by 4.25º (~250 km), but in this study, we only consider measurements from the center (nadir) slit, as this is the data that is currently released to the public (Kramarova 2023). The first OMPS LP was launched onboard the SNPP satellite in October 2011 and began operational observations in April 2012.

OMPS LP ozone profile retrievals are described in Rault and Loughman (2013) and Kramarova et al. (2018). Recently the retrieval algorithm was updated from version 2.5 to version 2.6. Several incremental improvements in calibration, retrieval algorithm and data quality were made for OMPS LP version 2.6 ozone profile retrievals over version 2.5, including combining the UV and visible channels into a single retrieval, as detailed in Kramarova et al. (2024). A filter was also introduced to remove profiles affected by the Hunga Tonga eruption in 2022-2023. This filter is based on retrieved aerosol extinction and results in gaps in OMPS LP ozone observations in the lower stratosphere (12.5-22.5 km) in the southern midlatitudes and tropics (45°S–20°N) that persist for several months after the eruption.

Validation of version 2.5 showed mean differences with correlative measurements of less than ±10% between 18 and 42 km, with a negative bias above 43 km and larger biases in the lower stratosphere and upper troposphere; there was also a positive drift of ~0.5%/yr which is more pronounced above 35 km (Kramarova et al., 2018). Comparisons of version 2.6 retrievals with Aura MLS by Kramarova et al. (2024) found that the algorithm improvements have helped to reduce vertical oscillation seen in version 2.5 and negative biases above 45 km have been reduced. Mean biases compared to MLS are within ±10% above 20 km and in many places less than ±5%; there has also been a reduction in the relative drifts between OMPS LP and MLS to less than 0.2%/yr in the upper stratosphere above 35 km (Kramarova et al., 2024).

This study focuses on the validation of OMPS LP version 2.6 ozone profile retrievals for the period April 2012 to April 2024. All OMPS LP data have been filtered using the suggested quality flags as described in the dataset readme document (Kramarova & DeLand, 2023).

## 3.   Correlative satellite and ground-based datasets





SNPP OMPS LP version 2.6 profiles have previously been compared to MLS (Kramarova et al., 2024). However,
MLS will be decommissioned in the coming year, so we also need to investigate alternative sources of correlative data
with which to validate OMPS LP ozone profiles. Ozonesonde observations offer one such dataset, however the
geographical and temporal extent of the data is limited. Other satellite data are available, and although solar occultation
instruments such as ACE-FTS and SAGE III/ISS may not provide such extensive spatial coverage as MLS, they are
able to provide high vertical resolution ozone profiles at different latitude bands throughout the year, providing the
opportunity for near global seasonal validation of OMPS LP ozone profiles.

**3.1. Ozonesondes**
Ozonesondes provide high accuracy, in situ, ozone profile observations from the surface up to approximately 30 km
altitude, however the data are spatiotemporally sparse. In this study we use data from 31 ozonesondes sites distributed
throughout the globe; Figure 1 shows a map of sites used and table S1 lists the site names, data sources, principal
investigator names and affiliations. Ozonesonde sites were selected for use based on continuity of data for the OMPS
LP measurement evaluation period of April 2012 to June 2024. A recent study by Stauffer et al., (2022), which
compared data from a network of 60 ozonesonde stations with satellite data, showed that when compared to Aura
OMI, total column ozone was stable to within about ±2% over an 18 year period, with similar results when compared
to three other total column satellite instruments. When compared to MLS, stratospheric ozone from sondes agreed to
within ±5% from 50 to 10 hPa. The study concluded that overall, global ozonesondes network data are of high quality
and stability.

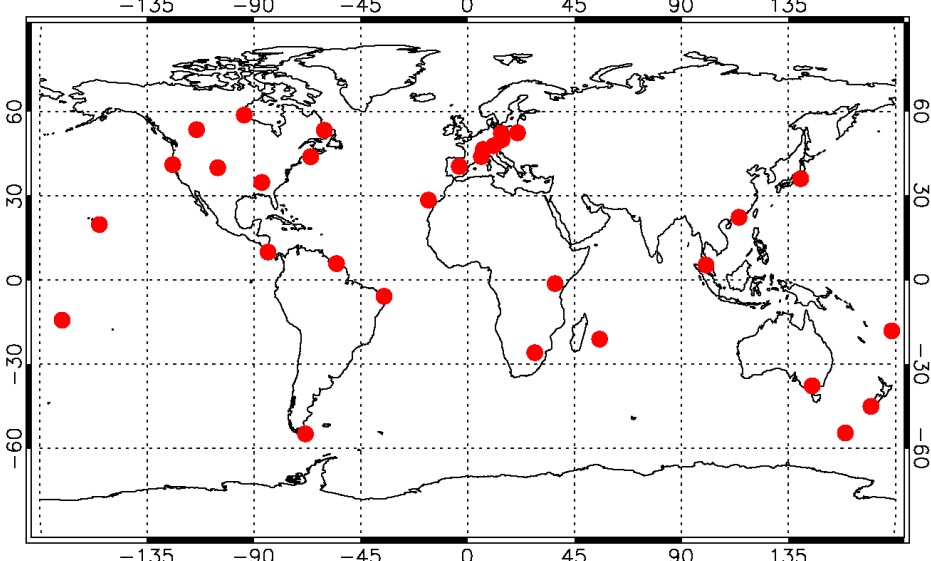

**Figure 1: Location of ozonesondes sites used for validation of SNPP OMPS LP version 2.6 ozone retrievals.**

**3.2. ACE-FTS**
The Atmospheric Chemistry Experiment-Fourier Transform Spectrometer (ACE-FTS), is a solar occultation satellite
instrument that makes measurements of ozone and other trace gases at sunrise and sunset (Bernath et al., 2005;
Bernath, 2017). ACE-FTS was launched onboard the Canadian Space Agency's SCISAT-1 satellite in 2003 and
therefore provides correlative data for the entire SNPP OMPS LP record. In this study we use ACE-FTS data version
5.2 (Bernath et al., 2025) and apply the quality flags of Sheese & Walker (2023). Since ACE-FTS only measures at
sunset and sunrise, and its orbit is optimized to provide coverage over polar mid and high latitudes, there are a limited
number of co-located profiles with which to compare with global OMPS LP observations, see Fig. 2. ACE-FTS



version 5.2 ozone retrievals have been validated against ozonesonde observations in a study by Zuo et al. (2024).
These results show that ACE-FTS ozone profiles have a general high bias in the stratosphere increasing with altitude
up to ~10% at ~30 km, with generally small insignificant drifts in the stratosphere (0-3%/decade). Comparisons with
ozonesondes only extend up to ~30 km, for higher altitudes, only previous versions have been validated against other
satellite instruments. Validation of ACE-FTS version 4.1 profiles shows that ACE-FTS ozone has a positive bias of
2-9% in the middle stratosphere that is stable to ±0.5%/decade, and a positive bias in the upper stratosphere that
increases with altitude up to ~15% and is stable to within ±1%/decade (Sheese et al., 2022). The estimated precision
for version 4.1 ozone retrievals is on the order of ~5-10% (Sheese et al., 2022).

**Figure 2: Co-located SNPP OMPS LP and ACE-FTS observations by month for the period 2012-2024.**

## 3.3. SAGE III/ISS

Like ACE-FTS, the Stratospheric Aerosol and Gas Experiment (SAGE) III, is a solar occultation instrument that
makes measurements of ozone profiles at sunrise and sunset (Cisewski et al., 2014). SAGE III/ISS was docked to the
International Space Station (ISS) in 2017 and began collecting data in June, thus providing nearly 8 years of correlative
data to compare with OMPS LP. In this study we use SAGE III/ISS ozone data version 6.0 (SAGE III/ISS data product
user's guide, 2025). Owing to the fact that SAGE III/ISS is a solar occultation instrument and is on board the ISS, it
provides limited global coverage which varies seasonally, doesn't extend north/south of 60 degrees latitude, and has
more frequent sampling of the tropics. Therefore, as with ACE-FTS there are a limited number of co-located global
profiles with which to compare with OMPS LP, see Fig. 3. The latest version of SAGE III/ISS ozone to be validated
was v5.1 (Wang et al., 2020). Those results showed that SAGE III/ISS ozone has a precision of ~3% in the 20-40 km
altitude range which degrades due to lower signal-to-noise ratios at higher and lower altitudes, reaching ~10-15% in
the upper stratosphere/lower mesosphere (~55 km) and ~20-30% near the tropopause. The mean biases when
compared to ozonesondes, lidars and other satellite correlative measurements are less than 5% for ~15-55 km in the
mid-latitudes and ~20-55 km in the tropics, increasing to 10% near the tropopause and to 15% at 60 km. Subsequent
changes applied in version 5.3 to the ozone retrievals have led to degraded precision (5% in the mid/lower
stratosphere), but increased vertical resolution, a reduction in low-altitude biases and a slight reduction in random





noise. Changes made to version 6 have led to an increase in retrieved ozone of around 3% due to switching to the new
ozone absorption coefficients (SAGE III/ISS data product user's guide, 2025).

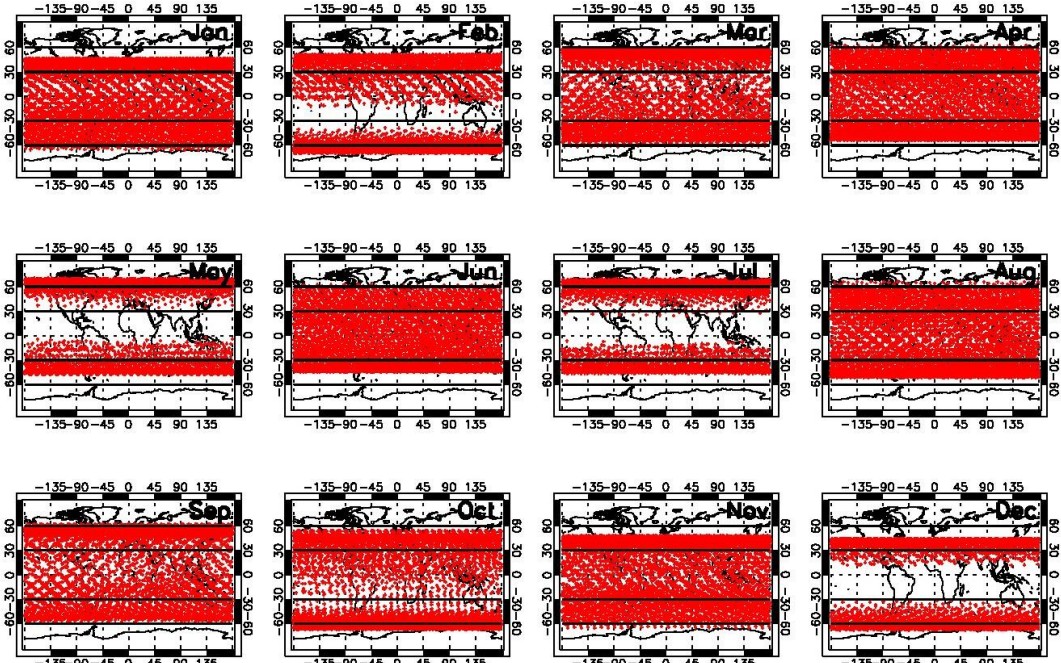

**Figure 3: Co-located SNPP OMPS LP and SAGE III/ISS observations by month for the period 2017-2024.**

**3.4.  MLS**
The Microwave Limb Sounder (MLS, Waters et al., 2006) provides global profile observations of ozone and other
trace species. MLS was launched on board the Aura satellite in 2004 and so provides correlative data for the entire
OMPS LP record to date. In this study we use version 5 of MLS data (Livesey et al., 2022). Since OMPS LP only
measures during the day, we only use MLS daytime observations, we also filter MLS data using criteria recommended
by the MLS Team. Both SNPP and Aura are in similar orbits with very similar equator crossing times and so MLS
provides excellent co-located profiles for global comparisons with OMPS LP. MLS ozone profiles have a precision
of 2-4% in the 18-43 km altitude range and this rapidly degrades outside of this altitude range (Livesey et al., 2022).
The accuracy of MLS ozone profiles ranges from 5 to 10% in the 12-57 km altitude range (Livesey et al., 2022), which
is the altitude range of interest in this study. Comparisons of MLS ozone using satellite, balloon, aircraft and ground-
based data have indicated general agreement at around 5-10% (Livesey et al., 2022). MLS exhibits drifts with respect
to ground-based networks of 1.5-2%/decade but with zero drift encompassed by the error bars, at least in the middle
stratosphere, and so is not statistically significant (Livesey et al., 2022).
**4.  Comparison Methodology**
In this study we apply common coincidence criteria to all correlative data to match OMPS LP profile sampling. Our
spatial coincidence criteria require profiles to be within ±2º latitude and less than 1000 km distance from the OMPS
profile. In order to maximise the number of comparison profiles, the only time criterion is that the profiles be on the
same day. If more than one profile matches these criteria then the spatially closest profile is used. We analyse all
profiles on the native OMPS LP coordinate system (number density/altitude), and do not account for the small



differences in the vertical resolution of the different measurement systems. Both MLS and ACE report ozone
concentrations in volume mixing ratio, in order to convert this to number density for comparison to OMPS we need
temperature and pressure information. For MLS, we use GEOS-FPIT temperature and pressure, and for ACE we use
temperature and pressure retrieved by ACE itself. No transformation is needed for SAGE III/ISS or ozonesonde data
as these data are provided as ozone number density profiles, however these data are provided on different altitude
grids to OMPS LP. Ozonesonde data are converted, where necessary, from volume mixing ratio to number density,
and from pressure grids to altitude grids, using the pressures and temperatures reported in the original data files, these
are then transformed onto the OMPS LP vertical grid via log-linear interpolation. SAGE III/ISS data are provided on
a 0.5 km vertical grid and so no interpolation is needed, we simply select matching altitudes for comparison profiles.
Stratospheric ozone exhibits diurnal variability, particularly above 30 km, which is both seasonally and latitudinally
dependent. The OMPS LP is a solar scattering instrument with a 1:30 pm equatorial crossing time that makes
observations in the sunlit portion of the Earth, whereas both ACE-FTS and SAGE III/ISS are solar occultation
instruments that measure ozone only at sunrise and sunset. We must therefore take into account the effects of any
diurnal changes in ozone between the OMPS LP observations and those of ACE-FTS and SAGE III/ISS. This is
achieved through the use of the Goddard Diurnal Ozone Climatology (GDOC) which is used to adjust both ACE-FTS
and SAGE III/ISS observations to the 1:30 pm local solar time - the measurement time of OMPS LP. Diurnal
adjustment using this climatology has been shown to reduce biases above 30 km (Frith *et al.* 2020).
Initially matched profiles were averaged into 5 degree zonal means for comparison, but owing to limited data coverage
from correlative solar occultation satellite observations (see Figs. 2 & 3), the data were further averaged into 3 wide
latitude bands to increase the number of observations in each bin for comparison statistics. These bands are 30ºS-60ºS,
30ºS-30ºN and 30ºN-60ºN and exclude the polar regions.

## 211 5. Results

### 212 5.1. Global profile comparisons

The mean biases for SNPP OMPS LP ozone retrievals compared to matched correlative measurements are shown in
Fig. 4. The upper panels a-c of figure 1 show zonal mean biases (5º latitude bins) between OMPS LP, ACE-FTS and
MLS as a function of altitude for the period 2012-2024 (2017-2024 for SAGE III/ISS). Panels d-f of figure 1 show
the mean biases for OMPS LP compared to all correlative measurement sources (ACE-FTS, SAGE III/ISS, MLS and
ozonesondes) as a function of altitude for 3 wide latitude bands, averaged over the period 2012-2024 (except SAGE
III/ISS, which is 2017-2024). The standard error of the mean for each comparison is also shown as horizontal bars,
standard deviations for these comparisons are shown in Fig. S1 in the supplemental material. SNPP OMPS LP version
2.6 ozone shows very good agreement with all correlative data sources between ~20 and 50 km at all latitudes, with
differences of less than ±10%, and between 20 and 45 km the differences between OMPS and MLS, and OMPS and
SAGE III/ISS are less than 5% in the tropics and southern mid-latitudes. Above 50 km, at all latitudes, the agreement
is still on the order of 10% or better, but differences with SAGE III/ISS and ACE-FTS start to increase with increasing
altitude above 55 km. This is consistent with the SAGE III/ISS and ACE-FTS validation results which show that both
instruments have an increasing positive bias in the upper stratosphere (Wang et al., 2020 and Sheese et al., 2022). It
is worth noting that without applying a diurnal correction to the ACE-FTS and SAGE III/ISS data the biases relative
to these datasets are even larger by up to 10%.
Below 20 km, the agreement between OMPS LP and correlative measurements varies by latitude, with larger positive
biases in the Upper Troposphere Lower Stratosphere (UTLS) layer (~15 to 20 km) between approximately 40º South
and 40º North. In the southern mid-latitudes OMPS LP agrees to within ~12% between 12 and 20 km when compared
to ACE-FTS, MLS and sondes, but shows slightly larger differences with SAGE III/ISS below 15 km. Below 20 km
in the northern mid-latitudes, the biases between OMPS LP and all correlative measurements are comparable, and
range from a positive bias of ~10% at 18 km down to a small negative bias of <5% at 12 km.
Overall, SNPP OMPS LP version 2.6 ozone profile biases do exhibit some vertical structure, with negative biases in
the lowest part of the profile (<15 km), followed by a positive bias up to ~20 km, then a negative bias again up to ~32
km, then a positive bias up to 40 km and then negative again above 40 km. This vertical pattern is somewhat latitude



dependent, with the low altitude negative bias being stronger in the tropics and southern hemisphere, and the positive
bias observed between ~32 and 40 km not present at latitudes north of 40ºN. However, almost all the biases when
compared to correlative data from other satellite instruments (ACE-FTS, SAGE III/ISS and MLS) fall within the
reported biases and precisions of those instruments.
Figure 5 shows vertical profiles of correlation coefficients between OMPS LP and matched correlative observations
for 3 wide latitude bands. In the mid-latitudes correlations of approximately 0.9 are seen between OMPS LP and ACE-
FTS and OMPS LP and MLS at most altitudes, and approximately 0.8 between OMPS LP and SAGE III/ISS between
15 and 40 km. Above 40 km the correlation with SAGE III/ISS drops rapidly reaching less than 0.5 at around 50 km,
indicating a spread in the biases at higher altitudes, this is also evident in the standard deviations of the profile
comparisons shown in Fig. S1. This is consistent with degraded precision and increased noise for SAGE III/ISS
measurements above 40 km as noted by Wang et al., (2020).
In the tropics, correlations between OMPS LP and MLS are around 0.8 up to 45 km dropping with increasing altitude
to 0.5 at 57 km, correlations with SAGE III/ISS are approximately 0.8 up to 37 km and then drop with increasing
altitude to 0.1 at 57 km. Correlations with ACE-FTS are between 0.4 and 0.8 throughout the entire vertical range with
a stronger correlation below 25 km. The drop in correlations seen at around 25 km at all latitudes and against all
correlative sources is likely due to the fact that this is where the peak in ozone density and  variability  is lower which
leads to weaker correlations. The correlations between OMPS LP and MLS and OMPS LP and ACE-FTS are improved
at all altitudes and latitudes for version 2.6 over version 2.5, with the largest improvement seen in the tropical lower
stratosphere where correlations between version 2.6 and MLS and ACE-FTS are greater than 0.8, whereas version 2.5
correlations were less than 0.7 compared to ACE-FTS and peaked at 0.8 compared to MLS (Kramarova et al., 2018).



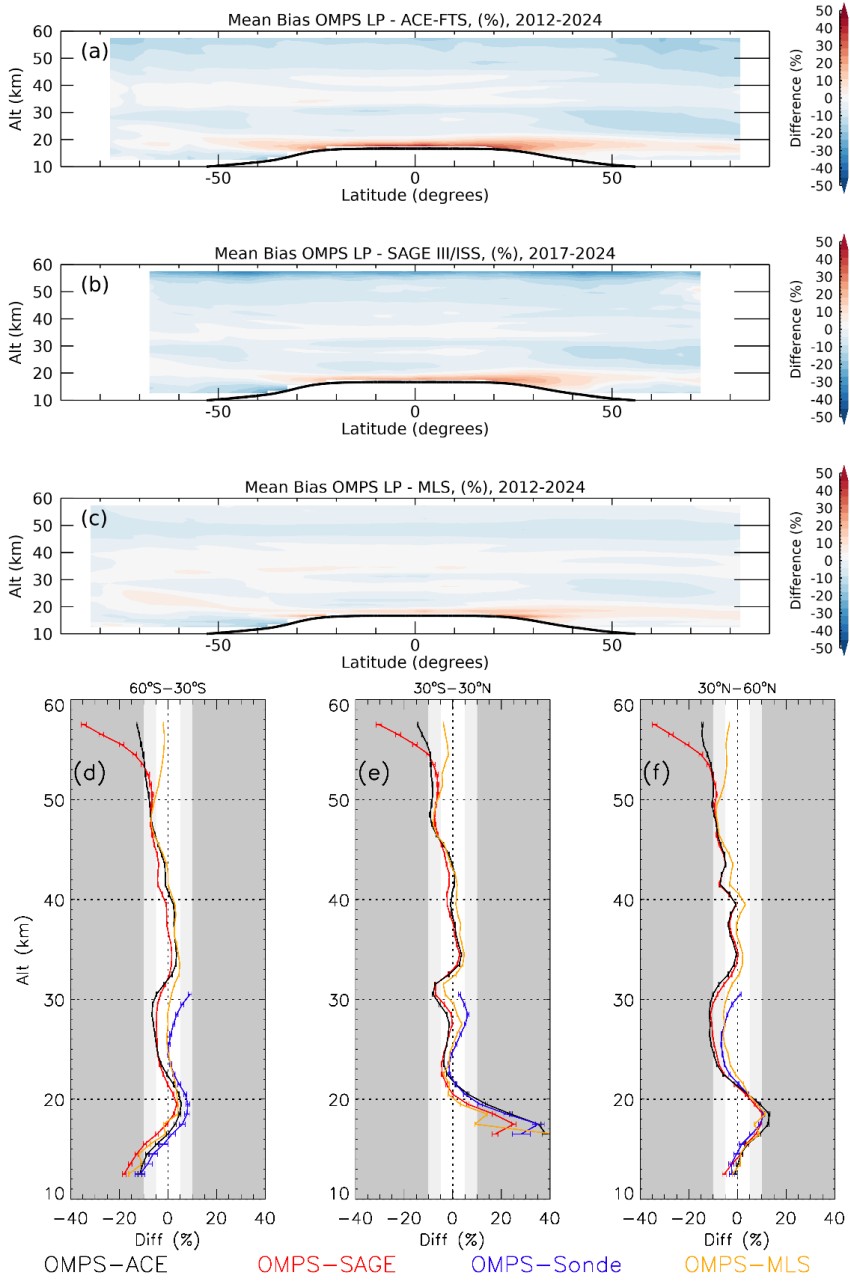

**Figure 4: Profile differences between OMPS LP and matched correlative satellite and ground-based observations. Panels (a-c) show zonal mean differences between OMPS LP and ACE-FTS, OMPS LP and SAGE III/ISS and OMPS LP and MLS on a 5° latitude grid. Panels (d-f) show mean profile differences between OMPS LP and ACE-FTS (black), OMPS LP and SAGE III/ISS, OMPS LP and ozonesondes (blue) and OMPS LP and MLS (Orange) for 3 wide latitude bands, the horizontal bars show 2 times the standard error of the mean (SEM), the white area indicates differences less than 5%, the light grey area 5-10% and the dark grey area represents differences greater than 10%, only data above the tropopause are shown.**





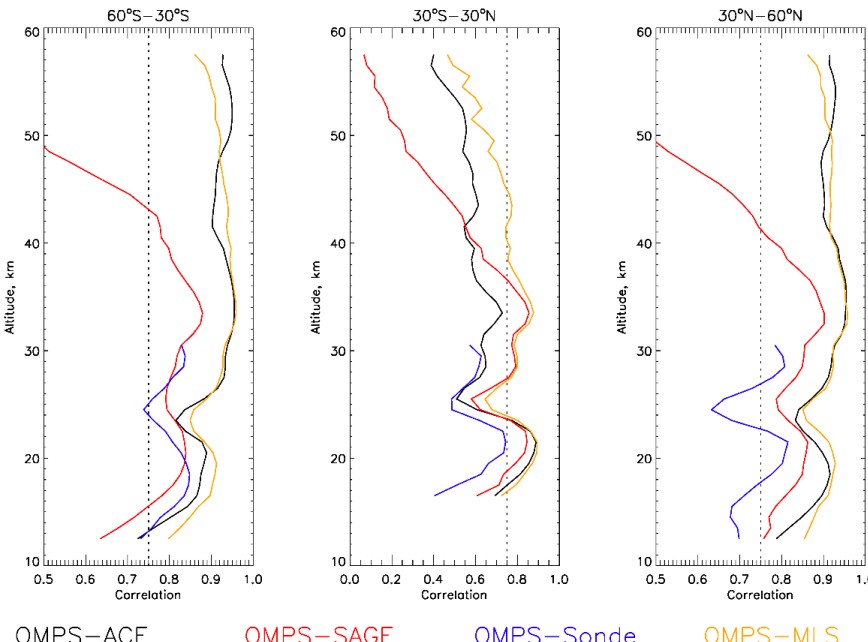

**Figure 5: Vertical profiles of correlation coefficients between OMPS LP and matched correlative observations for 3 wide**
**latitude bands.**

## 5.2. Seasonal cycle

To evaluate how well OMPS LP captures the seasonal cycle in ozone we compare the ozone seasonal cycle for each
correlative dataset to co-located OMPS LP observations in 3 wide latitude bands as used previously. The seasonal
cycle is calculated by taking each set of co-located OMPS LP and correlative data and subtracting the long-term mean
from the monthly mean (for all years) for each latitude band. Figure 6 shows seasonal cycle comparisons between
OMPS LP and all correlative measurements (ACE-FTS, SAGE III/ISS, MLS and sondes) at 4 altitudes (20,30, 40 and
50 km), the dashed lines represent the correlative observations seasonal cycles and the solid lines represent the co-
located OMPS LP seasonal cycles. The shape of the seasonal cycle is generally consistent between OMPS LP and all
3 correlative observation sources at all altitudes and latitudes. The seasonal cycle seen in ACE-FTS differs from the
other instruments in the Southern Hemisphere at 30 km and above, this is likely due to the differences in sampling
between ACE-FTS and the other instruments (see figures S2 and S3) as the OMPS LP co-located seasonal cycle has
also changed from those for the dense coverage satellites (e.g., OMPS-MLS matches).
Figure 7 shows the seasonal cycle biases between OMPS LP and the correlative datasets (difference between solid
and dashed lines in Fig. 6). There are small biases evident between the OMPS LP seasonal cycles and the seasonal
cycles of correlate observations that vary by altitude and latitude, generally the biases are larger and noisier compared
to ACE-FTS and SAGE III/ISS and are smaller and smoother compared to MLS, which may be a consequence of
sampling differences. Below 20 km in the mid-latitudes there is a pattern to the seasonal biases that is consistent across
all correlative datasets, with a high bias seen in the early part of the year (January-March), followed by a negative bias
in the middle of the year (April-September) and then a positive bias towards the end of the year (October-December).
Despite the pattern, although not the magnitude, of these biases being consistent across all correlative sources they
are, however not statistically significant, as indicated by the absence of black contour lines in Fig. 7. At 30 km a
consistent small positive bias is seen between April and September in the northern mid-latitudes when compared to
all correlative sources that is not present at other latitudes, this bias is significantly smaller than a similar bias observed




in OMPS LP version 2.5 which was attributed to an unexpected thermal sensitivity issue with OMPS LP (Kramarova
et al., 2018; Jaross et al., 2014). However, in version 2.6 this bias is not statistically significant. Above 50 km larger
biases are seen relative to ACE-FTS and SAGE III/ISS in the mid-latitudes with negative biases observed in the
spring/summer months and positive biases in the fall/winter months, some of which are statistically significant as
indicated by the black contours in Fig. 7. However, these biases are not seen when compared to MLS. These results
show that there are no significant biases in the OMPS LP seasonal cycle.

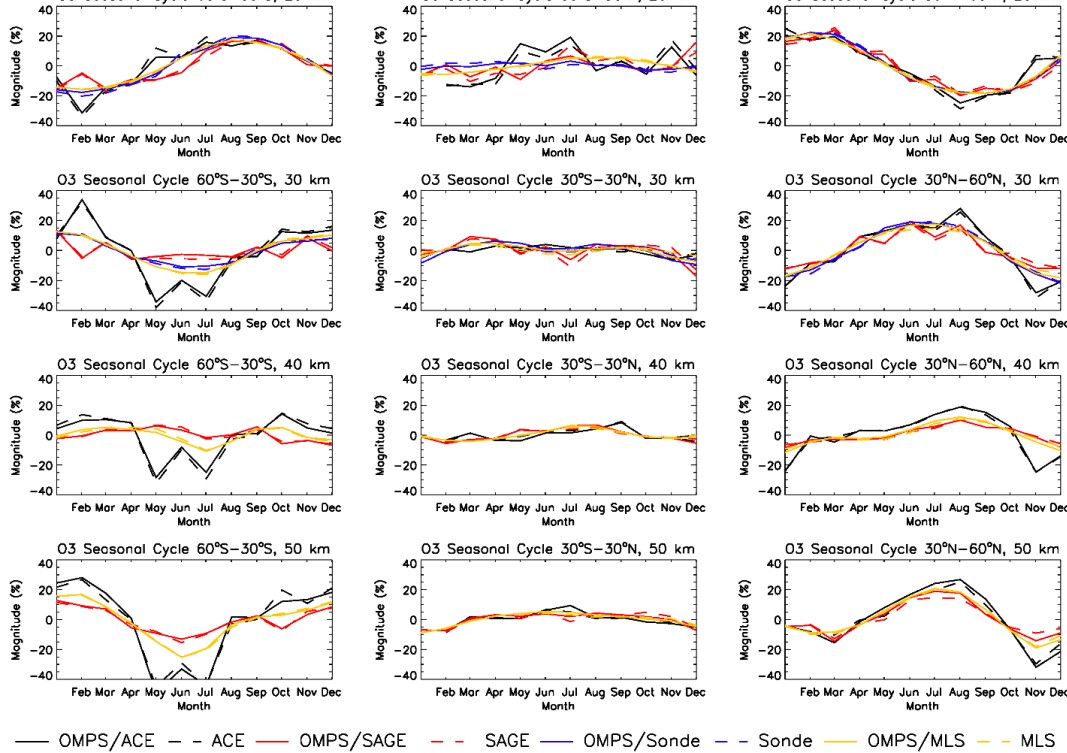

**Figure 6: Seasonal cycle in co-located OMPS LP (solid lines) and correlative observations (dashed lines) calculated as**
**monthly mean deviations from the long-term annual mean in % calculated for each instrument independently. OMPS**
**seasonal cycles are calculated using a sub-set of matching profiles for each correlative instrument.**



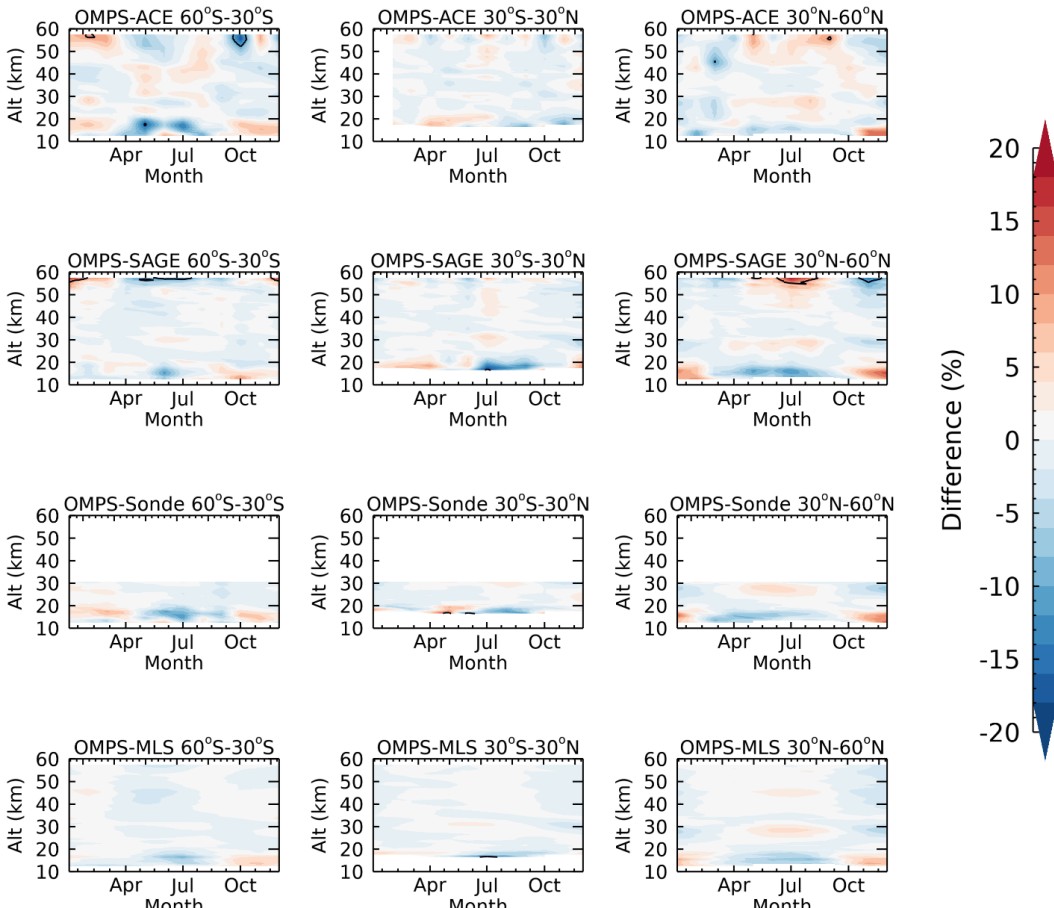

**Figure 7: Seasonal cycle biases between OMPS LP and correlative observations, calculated as the differences between the co-located OMPS LP seasonal cycle and the correlative observation seasonal cycle, black contours encompass biases that are larger than 2 standard deviations.**

### 5.3. Long term stability of OMPS LP ozone

In order to assess the stability of OMPS LP version 2.6 ozone retrievals over time we calculate their drift with respect to correlative measurements. Drifts are determined by calculating a linear fit for monthly mean deseasonalized co-located differences between OMPS LP and each correlative dataset within each latitude band. Figure 8 shows the calculated drifts in OMPS LP version 2.6 ozone relative to correlative measurements as a function of altitude above the tropopause; the shaded areas represent 2 sigma for the linear fit.

Between 25 and 50 km OMPS LP exhibits a small drift over the 2012-2024 period relative to MLS (less than 0.2 %/yr) which is positive in the tropics and northern mid-latitudes (Fig. 8b-c), and negative in the southern mid-latitudes (Fig. 8a). The drifts relative to ACE-FTS and SAGE III/ISS at these altitudes remain predominantly negative at all latitude bands and rise from less than -0.1 %/year at 25 km to -0.3 %/year at 50 km (Fig. 8a-c), except for the tropics where SAGE III/ISS has a much larger drift at 50 km (-0.8 %/yr) than the other data sources (Fig. 8b), this is due to the shorter time period where SAGE III/ISS and OMPS overlap (see discussion below). The drift relative to sondes appears consistent with MLS and ACE-FTS in the mid-latitudes (Fig. 8a&c), but diverges in the tropics between 22





and 20 km (Fig. 8b), exhibiting a positive relative drift of up to +0.2 %/yr whereas the satellite observations show a
small negative drift of up to -0.2 %/yr. The small drifts of opposing signs observed between the different data sources
indicate that OMPS LP exhibits no significant systematic drift between 25 and 50 km for the period 2012 to 2024.
Above 50 km, in the tropics and southern mid-latitudes (Fig. 8a-b), drifts relative to MLS and ACE-FTS remain small
(less than 0.2 %/year). In the northern mid-latitudes (Fig. 8c), the drift relative to MLS increases slightly and is positive
(up to +0.4 %/yr) whereas the drift relative to ACE-FTS is negative (up to -0.4 %/yr). Again, the fact that the drifts
relative to MLS and ACE-FTS are either close to or straddle the zero line, suggests that there is no significant
systematic drift in OMPS LP at these altitudes over the 2012-2024 period.
The eruption of the Hunga volcano in January 2022 caused problems for OMPS LP ozone retrievals because of high
aerosol loading at 25 km and below, leading to anomalously high ozone being reported. A filter based on aerosol
optical depth was implemented for OMPS LP ozone (Kramarova et al., 2024), this dramatically reduced the number
of OMPS LP observations at altitudes below 25 km in the tropical and southern mid-latitude regions in the months
following the eruption. However, even after this filter is applied, a higher than normal bias is still observed with
respect to correlative observations in these regions that persists throughout 2022 and well into 2023. This positive
anomaly is small when compared to MLS, but is larger when compared to ACE-FTS and is largest when compared to
SAGE III/ISS as shown in Fig. S2 that demonstrates the time series of differences over the 2020-2025 period. Both
SAGE III/ISS and ACE-FTS already had a limited number of observations in these latitude bands depending on the
season, and with the reduction of OMPS LP observations the remaining number of matches in the low stratosphere
for these two instruments is severely reduced from early 2022 to mid to late 2023 as shown in Fig. S3. The resulting
drifts relative to ACE-FTS and particularly SAGE III/ISS below 25 km when calculated up to April 2024 appear to
be erroneously large, especially in the tropics and southern mid-latitudes. For these reasons, for altitudes below 25
km, we will focus on drifts calculated up to the end of 2021 only, which can be found in panels (d-f) in Fig. 8.
Between 20 and 25 km OMPS LP exhibits only a small drift (<±0.3 %/yr) relative to ACE, MLS and sondes over the
period 2012-2021 (Fig. 8d-f), with the largest drifts seen at 25 km in the tropics relative to ACE (Fig. 8e) and at 20
km in the southern mid-latitudes relative to sondes (Fig. 8d). In the tropics and southern mid-latitudes (Fig. 8d-e) the
drifts relative to different data sources straddle the zero line indicating no systematic drift for the time period 2012-
2021, in the northern mid-latitudes the drifts are generally all very small and positive. Below 20 km, for the period
2012-2021, in the mid-latitudes (Fig. 8d&f) the drifts relative to all data sources shows the same structure and start
out positive (~+0.2%/yr), but then show an increasing negative trend with decreasing altitude which peaks at ~-
0.6%/yr at around 15 km before improving at the bottom of the profile (except for ACE-FTS in the northern
hemisphere), with larger drifts seen at lower altitudes in the northern hemisphere. In the tropics there is a large spread
in the drifts relative to the four different data sources. The drifts relative to ACE-FTS and MLS are similar and those
relative to SAGE III/ISS and sondes have larger errors at these altitudes. ACE-FTS, SAGE III/ISS and MLS all show
a negative drift whereas the drift relative to sondes is positive.
These results represent an improvement in the long-term stability of OMPS LP ozone retrievals for version 2.6 over
version 2.5, with a reduction in drifts at all altitudes, particularly in the upper stratosphere where version 2.5 exhibited
drifts of 0.5-1%/yr (Kramarova et al., 2018). The observed small drifts and the spread in drifts relative to different
correlative data sources indicates that there is no significant systematic drift in OMPS LP version 2.6.
The drifts relative to SAGE III/ISS have larger magnitudes, sigmas, and different vertical structures to those of other
correlative measurements, particularly above 50 km and below 25 km. This is due to the shorter time period available
for OMPS LP-SAGE III/ISS comparisons. Once recalculated, the drifts relative to ACE-FTS and MLS above 50 km
and below 25 km exhibit similar magnitudes and vertical structures to that of SAGE III/ISS (Fig. S5). Analysis of the
time series of differences between OMPS LP and MLS in the 30ºN-60ºN latitude band for several altitudes over the
time period 2012 to 2024 (Fig. S6) show low frequency changes in ozone differences. Because of this the drifts for
the periods 2012-2024 and 2017-2024 are quite different with mostly negative drifts for the period 2017-2024,
however when we estimate the drift for the whole time period of 2012-2024 the drifts are much smaller. These time-
dependent changes in LP ozone are being investigated by the OMPS LP team, who also see time dependent changes
in radiance residuals (differences between calculated and measured radiances) at wavelengths that are not used in the
ozone retrieval that coincide with observed changes in ozone. Investigation of this behavior in other LP slits (not
shown here) suggest that this is not related to a drift in altitude registration. One possible explanation under
investigation is a potential shift in wavelength registration.




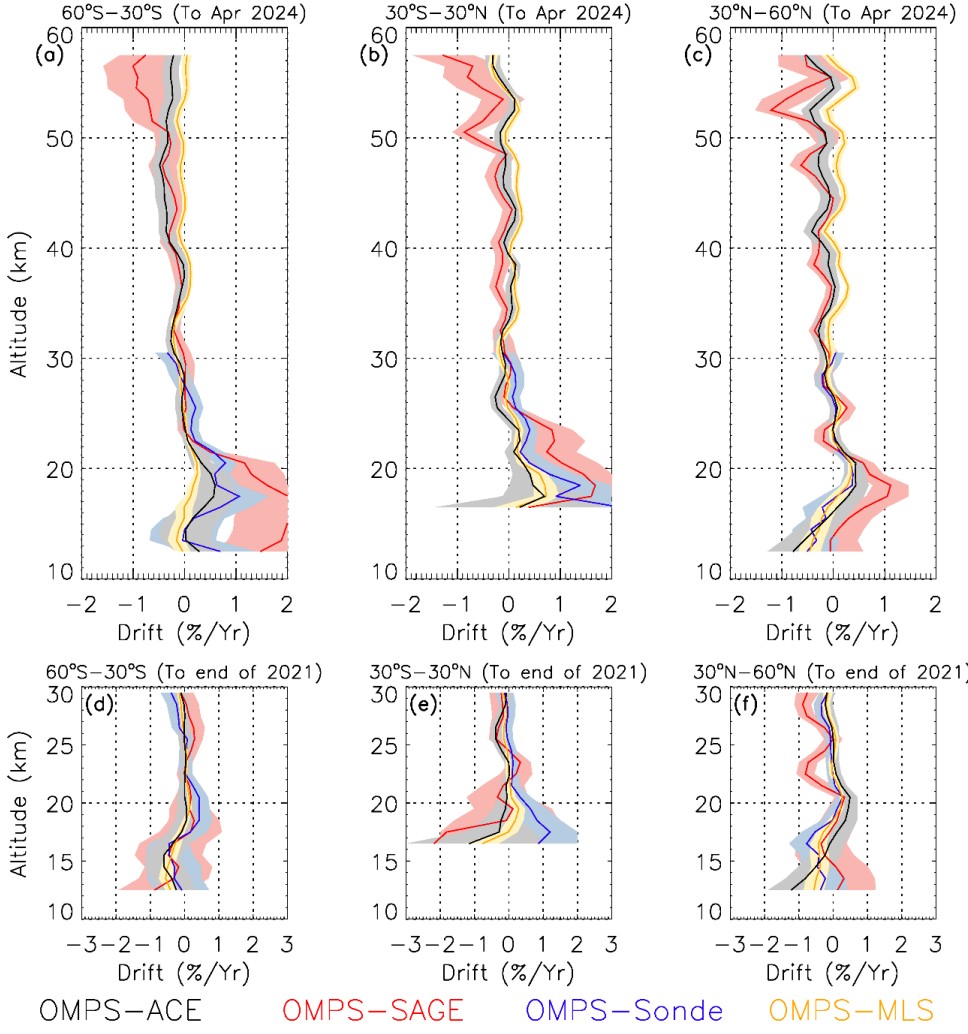

**Figure 8: Relative drifts for OMPS LP version 2.6 ozone in % per year relative to correlative observations, calculated using deseasonalized data from April 2012 to April 2024 for panels (a-c) and April 2012 to December 2021 for panels (d-f), except for SAGE III/ISS for which data starts in June 2017. Shaded areas show 2 sigma for the linear fit, only data above the tropopause is shown.**

## 6. Comparisons With Other Data Sources

In the future we won't be able to rely on having correlative satellite data with either both high vertical resolution and dense global sampling such as MLS, or high vertical resolution and limited global sampling such as SAGE III/ISS or ACE-FTS, as both MLS and SAGE III/ISS are scheduled to end operations in the near future and ACE-FTS is already long past its original planned mission lifetime. With no replacement missions for these instruments likely in the near future we will need to use other sources of correlative data with which to validate OMPS LP ozone retrievals in addition to ozonesondes. Here we investigate the use of lidar data and lower vertical resolution nadir satellite data.




### 6.1. Mauna Loa Lidar



One other source of high vertical resolution ozone profile measurements is ground-based lidar observations, of which
there are a limited number of stations located around the globe. Although the global coverage gained from lidar
observations is significantly lower than that of ozonesondes, lidars are able to observe ozone up to higher altitudes
than sondes (up to 50 km), therefore a combination of lidars and ozonesondes may provide a useful dataset for
validation of OMPS LP high vertical resolution ozone profile retrievals, albeit with limited global coverage. Here we
will compare to the Mauna Loa lidar station (MLO). The MLO lidar measures vertical ozone profiles from 15-50 km
at night, several times a week, with a vertical resolution of ~1 km near the ozone peak (~25 km) which decreases to
~3 km at the bottom of the profiles and to 8-10 km at the top of the profiles (Leblanc and McDermid, 2000). The
typical instrumental error is a few percent at the ozone peak and increases to 10-15% at ~15 km and to more than 40%
above 45 km (Leblanc and McDermid, 2000).

In this study we utilize MLO lidar ozone data for the period April 2012 to December 2022 to evaluate OMPS LP
version 2.6 ozone retrievals and compare these results to those of coincident MLS and ozonesonde comparisons to
OMPS LP at this location. Figure 9 shows mean profile differences between OMPS LP and MLO lidar data together
with collocated differences between OMPS LP and MLS and OMPS LP and ozonesondes launches from the Hilo
station. Between 20 and 45 km OMPS LP exhibits the same vertical structure in biases compared to both the MLO
lidar and MLS, with biases near zero between 20 and 25 km and between 40 and 45 km for both data sources. Between
25 and 40 km the bias compared to the MLO lidar (~5-10%) is larger than that with MLS (<5%), however between
23 and 30 km the biases with MLO and ozonesondes agree almost perfectly. Below 20 km the bias compared to the
MLO lidar is less than 10%, which is much smaller than the bias compared to both MLS and ozonesondes, however
the standard deviation of the MLO biases increases dramatically at these altitudes (see Fig. S4), likely as a result of
increased measurement error, and encompass the observed MLS and ozonesondes biases. Above 45 km, again the
MLO lidar and MLS biases differ, with the MLO biases being much smaller than MLS. This is also a region where
the MLO lidar measurement error increases dramatically and so does the standard deviation of the mean differences,
which again encompass the MLS biases.

Panel (b) of Fig. 9 shows a profile of the relative drift of OMPS LP compared to the MLO lidar. This is determined
by calculating a linear fit for monthly mean deseasonalized co-located differences for the time period of April 2012
up to the end of 2021. Between 20 and 40 km the drift in OMPS LP relative to the MLO lidar is very close to zero
(<±0.1%/yr), with the exception of a small positive drift (<0.4%/yr) between 20 and 24 km and a small negative drift
(<-0.3%/yr) at 32 km. Above 40 km the drift steadily increases with altitude reaching +1.2%/yr at 45 km and +1.8%/yr
at 50 km, however as previously noted the lidar measurement error increases dramatically above 40 km as does the
standard deviation of the differences, the vertical resolution of the lidar observations is also degraded to ~8-10 km at
these altitudes and so any observed trends in OMPS LP with fine vertical structure, natural or otherwise, would likely
lead to large drifts in the differences. Below 20 km drifts become increasingly negative increasing from ~0%/yr at 20
km to ~3%/yr at 16 km, again this is an altitude range with increased variability in the differences between the two
datasets and increased lidar measurement error.

The results are broadly consistent with MLS and sonde comparisons in the same location, although the existence of
some differences at higher and lower altitudes together with lidar observation error estimates, variability of differences
and changes in vertical resolution lead us to conclude that such data is most useful for evaluation of OMPS LP ozone
between 20 and 40 km. These results show that lidars can provide a useful dataset with which to evaluate OMPS LP
high vertical resolution ozone profile retrievals once MLS data is no longer available.


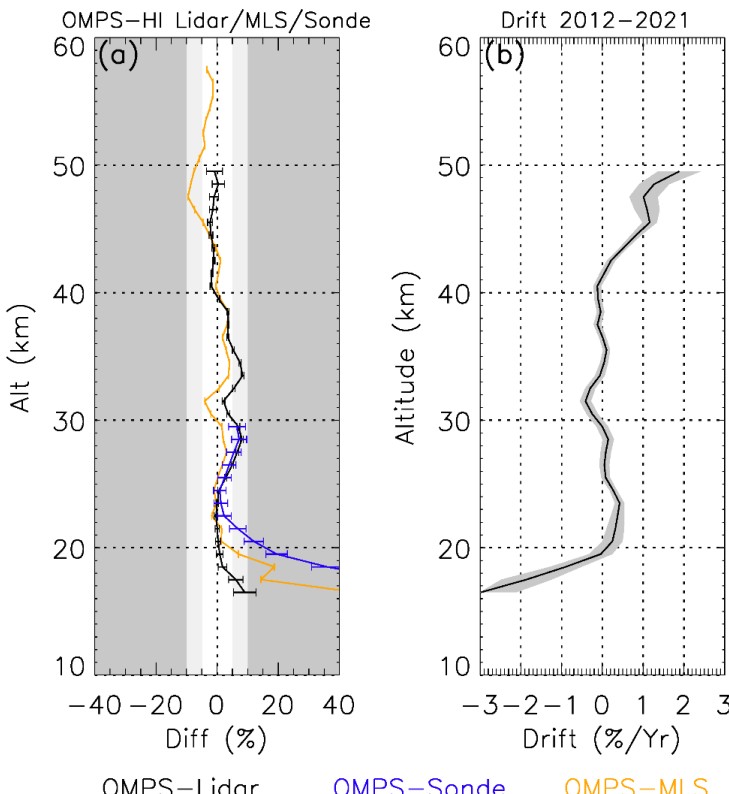

OMPS—Lidar    OMPS—Sonde    OMPS—MLS

**Figure 9: Mean profile differences and drifts between OMPS LP and Mauna Loa lidar observations. Panel (a) shows the mean profile differences between OMPS LP and lidar (black line), OMPS LP and MLS (yellow line), and OMPS LP and ozone sonde launches from Hilo (blue line), the horizontal bars show 2 times the standard error of the mean (SEM), the white area indicates differences less than 5%, the light grey area 5-10% and the dark grey area represents differences greater than 10%. Panel (b) shows the relative drift in % per year relative to lidar observations, calculated using deseasonalized data from 2012 to 2022. Shaded area shows 1 sigma for the linear fit, only data above the tropopause is shown.**

## 6.2. OMPS Nadir Profiler (NP)

The OMPS nadir profiler (OMPS NP) is a nadir viewing instrument that is part of the OMPS suite of instruments and measures vertical profiles of ozone (McPeters et al., 2019) with limited vertical resolution (6-8 km). Despite the limited vertical resolution, it has a number of advantages as a correlative data source for the evaluation of OMPS LP ozone profiles. It is on board the same spacecraft as OMPS LP, and so its observations are near coincident with LP observations in both space and time, it is able to provide the same global coverage as OMPS LP (every 3-4 days), although its profiles are of low vertical resolution they do cover the full vertical range of OMPS LP ozone retrievals, and there will always be an NP instrument as part of the OMPS  to provide data with which to compare. Ozone profile retrievals from SNPP OMPS NP have been demonstrated to agree with observations from NOAA-19 SBUV-2 to within ±3 % with an average bias of -1.1 % in the upper stratosphere and +1.1 % in the lower stratosphere (McPeters et al., 2019).





In this study we utilize SNPP-OMPS NP ozone profile for the period April 2012 to December 2024 to evaluate OMPS
LP version 2.6 ozone retrievals. In order to compare OMPS LP and OMPS NP, OMPS LP profiles were first converted
into partial ozone columns according to the OMP SNP pressure grid, the OMPS NP averaging kernels were then
applied to the OMPS LP profiles to degrade them to the OMPS NP vertical resolution. Figure 10 shows mean profile
differences between OMPS LP and OMPS NP averaged over the whole measurement time period for 3 wide latitude
bands. In general, the biases relative to OMPS NP are less than 5% at all altitudes and all locations, with the exception
of the tropical lower stratosphere (below 25 km). The biases for all locations show the same vertical oscillatory
structure which is stronger in the tropics and northern mid-latitudes. This manifests as positive biases below ~28 km,
negative biases between ~28 and ~36 km, positive biases between ~36 and ~46 km and negative biases above ~46 km
for these two regions. Also shown in Fig. 10 are the mean profile differences between MLS and OMPS NP for the
same latitude bands (yellow lines). Since the differences between MLS and OMPS NP exhibit the same oscillatory
vertical structure as the differences between OMPS LP and OMPS NP, we can conclude that this vertical structure is
an artifact of the OMPS NP measurements and not OMPS LP. With this in mind, the biases observed between OMPS
LP and OMPS NP are consistent with those seen between OMPS LP and other correlative observations.
Figure 11 shows the drift of OMPS LP relative to OMPS NP. This is determined by calculating a linear fit for monthly
mean deseasonalized differences for the time period of April 2012 up to the end of 2024 for altitudes above 25 km
and up to the end of 2021 for altitudes below 25 km. Between ~13 and ~35 km the drift relative to OMPS NP is small
<±0.1%/year in the mid-latitudes, and <±0.2%/yr in the tropics. Above 35 km the drift becomes positive in all latitude
bands and increases up to 0.3%/yr. These small drifts fall within the range of drifts seen when comparing OMPS LP
with other correlative measurements.
These results suggest that OMPS NP is able to provide a useful dataset with which to globally evaluate OMPS LP
ozone profiles, albeit with limited vertical fidelity. Bias calculations with OMPS NP introduce some oscillatory
vertical structures which are a characteristic of the OMPS NP measurements and not OMPS LP. This should be taken
into consideration when using OMPS NP to evaluate OMPS LP vertical profiles.

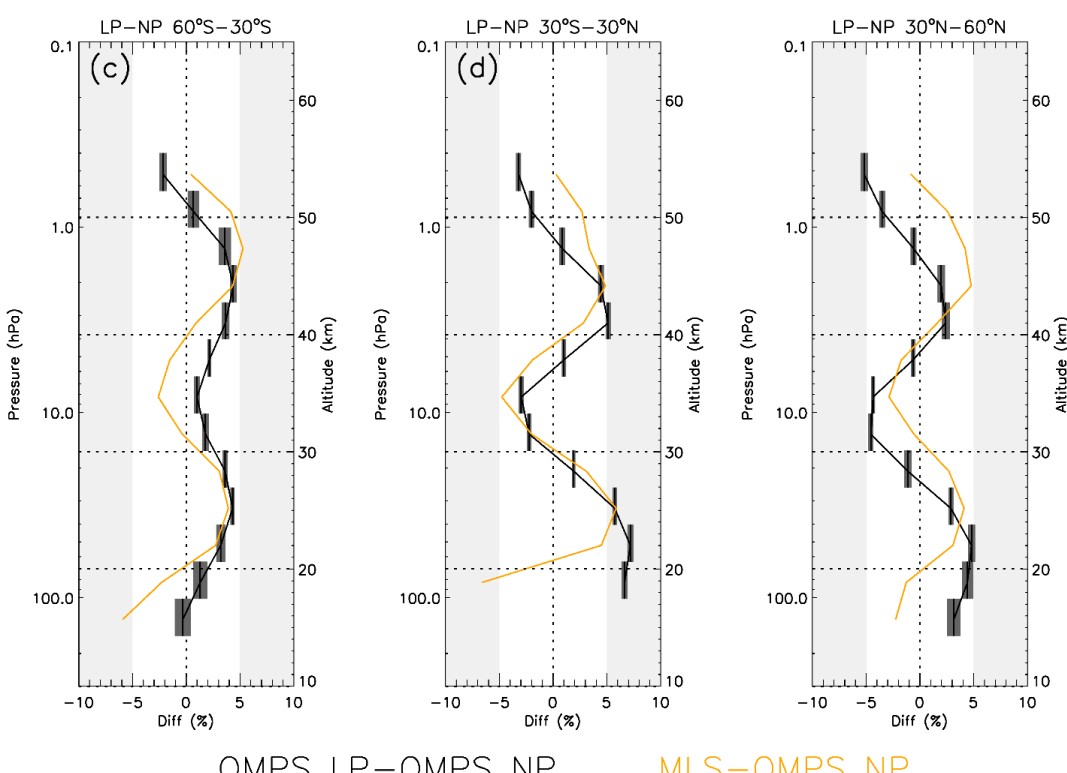

**Figure 10: Mean profile differences between OMPS LP and OMPS NP (black) for 3 wide latitude bands, the horizontal shading show 2 times the standard error of the mean (SEM), the white area indicates differences less than 5%, the light grey area 5-10%. Also shown in yellow are mean profile differences between MLS and OMPS-NP. Only data above the tropopause are shown.**



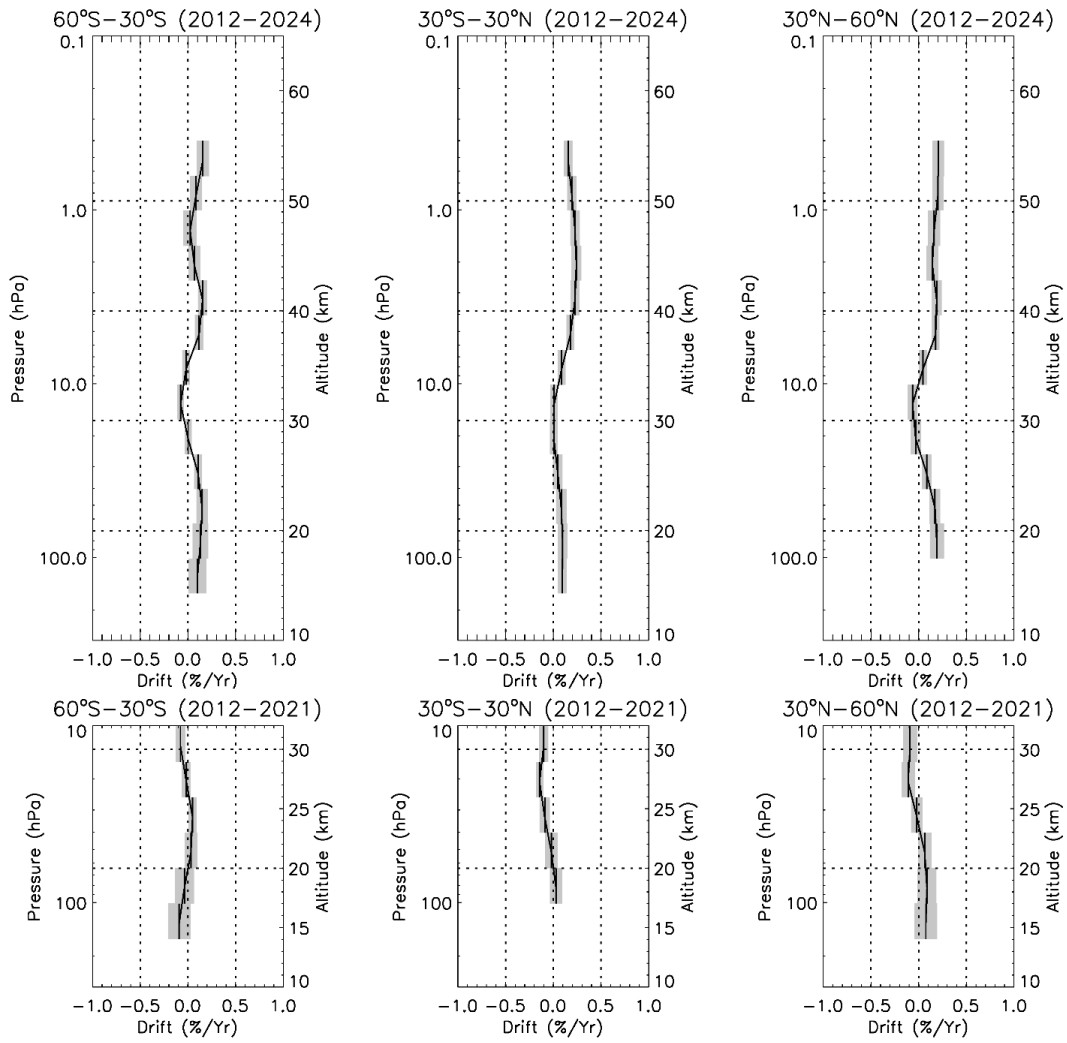

**Figure 11: Relative drifts for OMPS LP version 2.6 ozone in % per year relative to OMPS NP, calculated using deseasonalized data from April 2012 to December 2021. Shaded areas show 2 sigma for the linear fit, only data above the tropopause is shown.**

## 7. Comparisons in Polar Regions

Previously, we limited our comparisons geographically to exclude polar regions (latitudes greater than 60) owing to sparse data in this region from all correlative data sources. However, MLS and OMPS NP have sufficient data coverage that extends to latitudes greater than 60 degrees to evaluate OMPS LP ozone retrievals in these important regions.

Figure 12 shows mean profile differences between OMPS LP and correlative observations in two wide polar latitude bands, 82S-60S and 60N-82N. The biases relative to MLS are generally less than ±5% except between 45 and 55 km in the southern hemisphere where biases peak at -8% at 48 km, in the northern hemisphere there are 3 altitude regions





where the bias relative to MLS exceeds ±5%, 15-20 km, 23-30 km and 45-50 km, but those biases are still less than
±10%. Compared to OMPS NP, LP biases are generally less than ±5% except for approximately 25 to 30 km in the
southern hemisphere, and above ~52 km in the northern hemisphere.
Figure 13 shows relative drift profiles between OMPS LP and correlative observations for the two polar regions. The
drift relative to MLS is less than ±0.2%/yr at all altitudes in the polar regions except in the northern hemisphere at 18
km and between ~53 and 55 km. Drifts of less than ±0.3%/yr are seen relative to OMPS NP at all altitudes in both
polar regions, with drifts less than ±0.1%/yr above 25 km in the southern hemisphere, and less than ±0.2%/yr between
25 and 45 km in the northern hemisphere.

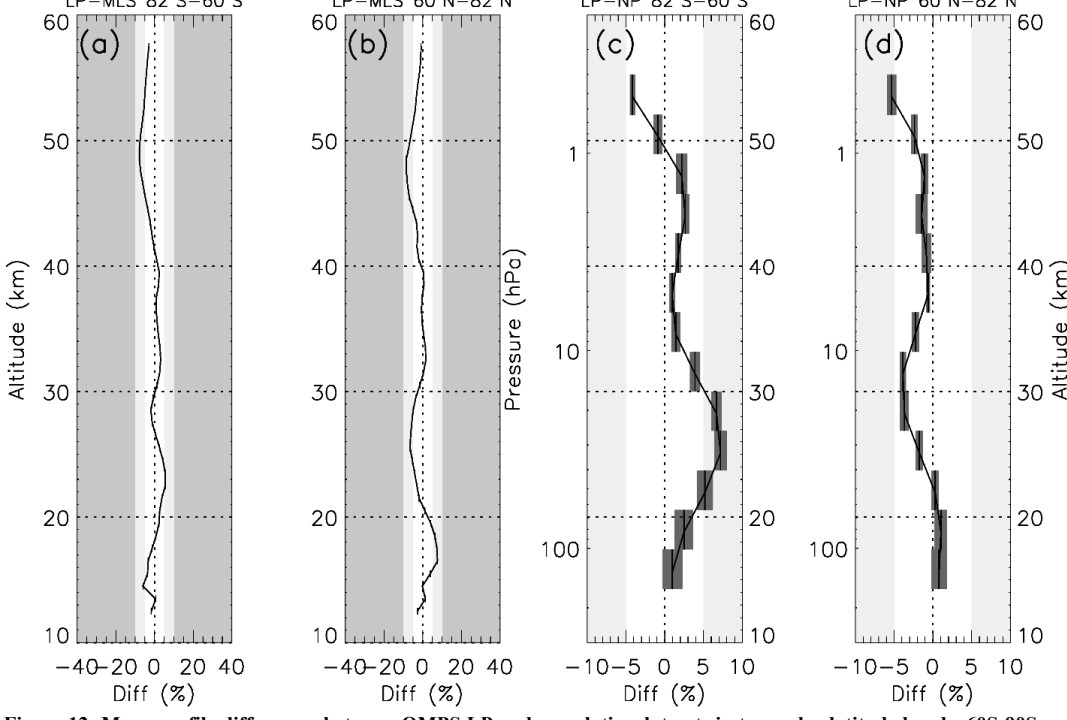

**Figure 12: Mean profile differences between OMPS LP and correlative datasets in two polar latitude bands, 60S-90S and**
**60N-90N. Panels (a) and (b) show the mean profile differences between OMPS LP and MLS, the horizontal bars show 2**
**times the standard error of the mean (SEM). Panels (c) and (d) show the mean profile differences between OMPS LP and**
**OMPS NP, the horizontal shading show 2 times the standard error of the mean (SEM). The white area indicates differences**
**less than 5%, the light grey area 5-10% and the dark grey area represents differences greater than 10%, only data above**
**the tropopause are shown.**





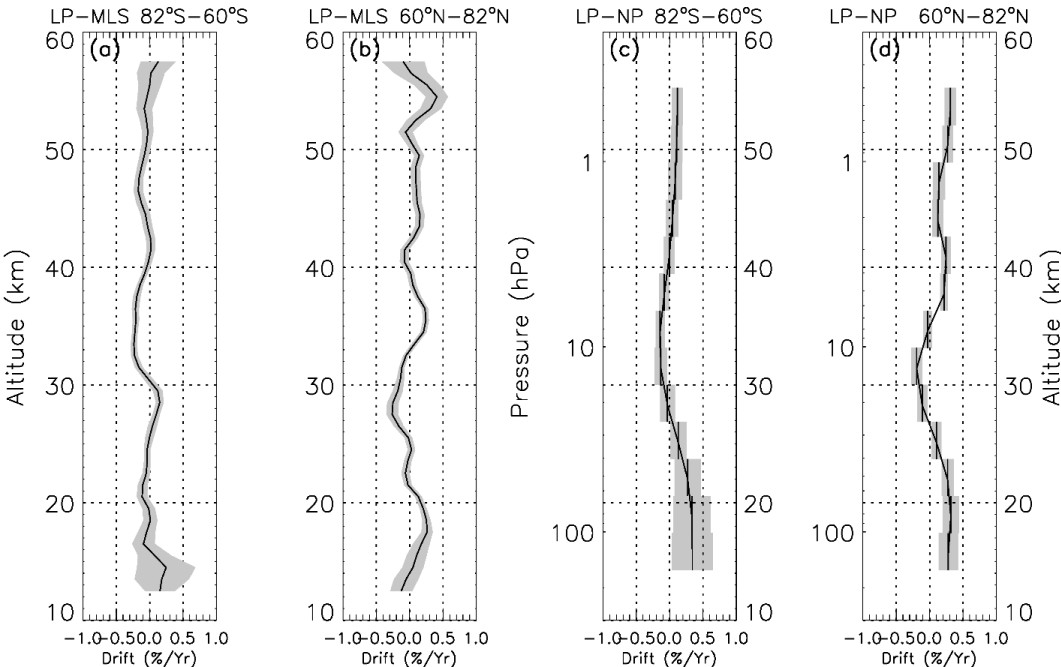

**Figure 13: Relative drifts for OMPS LP version 2.6 ozone in % per year, calculated using deseasonalized data from April 2012 to December 2024 for two polar latitude bands, 60S-90S and 60N-90N. Panels (a) and (b) show drifts relative to MLS. Panels (c) and (d) show drifts relative to OMPS NP. Shaded areas show 2 sigma for the linear fit, only data above the tropopause is shown.**

## 8. Discussion

Currently the best source of correlative data with which to evaluate OMPS LP ozone profile retrievals is Aura MLS as it is able to provide high vertical resolution profiles with dense geospatial sampling. However, MLS is scheduled to be decommissioned within the next year, and so other sources of data must be found. In this paper, in addition to MLS, we have used solar occultation satellite instrument data from ACE-FTS and SAGE III/ISS to evaluate OMPS LP ozone profiles. Although solar occultation instruments are able to provide high vertical resolution ozone profiles the number of profiles observed per day by these instruments is very small compared to OMPS LP, and their spatial coverage is very limited and varies seasonally. The limited number of observations and lack of spatial coverage means that in order to make meaningful global comparisons one must average over wide latitude bands and longer time scales. It also means that longer time periods are needed in order to calculate reliable drifts. Ozonesondes and lidar observations are able to provide high vertical resolution ozone profiles with which to evaluate OMPS LP profiles, however they lack the geospatial coverage afforded by a satellite instrument such as MLS. Together, independent solar occultation and ozonesonde measurements can be used to continuously monitor for potential drifts in OMPS LP, while LP provides the near global coverage necessary to ensure geographically representative trends. Finally, the OMPS NP series of instruments, which is able to provide full global coverage coincident with OMPS LP, but with limited vertical resolution, offers a source of data with which to evaluate OMPS LP ozone profiles, and there will always be an OMPS NP instrument on the same satellite platform as all future OMPS LP instruments. Therefore, in the future when MLS data is no longer available, a combination of ozonesondes, lidars and OMPS NPs will be needed in order to globally evaluate OMPS LP ozone profiles, with OMPS NP providing the global coverage and ozonesondes and lidars providing the high vertical resolution information needed to interpret any vertical structure seen in the OMPS LP/NP comparisons.



## 9. Conclusions

In mid-2023 a new version of OMPS LP ozone profile retrievals, version 2.6, was released. Version 2.6 includes a number of incremental improvements in calibration, the retrieval algorithm and data quality. In order to evaluate this latest version of OMPS LP ozone profile data, we compared OMPS LP version 2.6 ozone retrievals against correlative data from other satellite instruments (MLS, ACE-FTS and SAGE III/ISS) and ozonesondes for the time period 2012-2024 in three wide latitude bands from 60ºS to 60ºN.

Our results show very good agreement between OMPS LP and all correlative data sources between 20 and 50 km at all latitudes with differences of less than 10%, with OMPS generally exhibiting a negative bias, except between 32 and 38 km in the tropics and southern mid-latitudes, where the bias is positive. Between 20 and 45 km in the tropics and southern mid-latitudes the differences between OMPS LP and MLS, and OMPS LP and SAGE III/ISS are less than ±5%. Above 50 km, the agreement with MLS is still on the order of -5% or better, but differences with SAGE III/ISS and ACE-FTS start to increase with increasing altitude, which is consistent with the SAGE III/ISS and ACE-FTS validation results which show that both instruments have an increasing positive bias in the upper stratosphere. Below 20 km, larger positive biases, up to ~35%, are seen in the tropical tropopause layer (~15 to 20 km) between approximately 40º S and 40º N. In the southern mid-latitudes OMPS LP agrees to within ~12% between 12 and 20 km when compared to ACE-FTS and sondes, but shows slightly larger differences with MLS and SAGE III/ISS below 15 km. Below 20 km in the northern mid-latitudes, the biases between OMPS LP and all correlative measurements are comparable, and range from a positive bias of ~10% at 18 km down to a small negative bias of <5% at 12 km. Almost all of the observed biases when compared to correlative satellite data fall within the reported biases and precisions of those instruments, particularly in the 20 to 45 km altitude range.

We now have more than 12 years of OMPS LP ozone retrievals, and this allows us to evaluate both the seasonal cycle and the long-term stability of the data, which we have done by comparing to satellite and ozonesonde data. We find that OMPS version 2.6 ozone exhibits the same seasonal cycle as compared to all correlative measurement sources and our analysis shows that there is no significant seasonal bias in the OMPS LP.

To evaluate the long term stability of OMPS LP ozone we calculate the drifts between OMPS LP and correlative data sources using deseasonalized monthly mean differences. We find small drifts relative to all correlative observations at all latitude bands of less than ±0.2%/yr (±0.1%/yr) between 25 and 50 km, with larger drifts above 50 km (up to ±0.4%/yr) and below 20 km (up to ±0.6%/yr), these represent an improvement over OMPS LP version 2.5 ozone. However, there is a spread in these drifts between correlative sources that often straddles the zero line. We therefore conclude that there is no significant systematic drift in OMPS LP version 2.6 ozone for the period 2012 to 2024 and that it is suitable for use in ozone trend studies. Whilst relative drifts calculated over shorter time periods can be larger, as demonstrated here for the period 2017-2024, analysis of ozone difference time series does not show any clear, consistent, drifts in OMPS LP ozone over the entire record.

Finally, we investigated the possible use of lidar and OMPS NP observations for the future evaluation of OMPS LP ozone profile retrievals. We find that although neither data source is able to provide a suitable replacement dataset in terms of high vertical resolution and high geospatial global coverage alone, a combination of OMPS NP, ozonesondes and lidars would provide a reasonable dataset with which to evaluate OMPS LP ozone retrievals. OMPS NP would provide dense global sampling with near perfect colocation with OMPS LP, but with limited vertical resolution, whereas ozonesondes and lidars may have sparse spatiotemporal coverage but are able to provide high vertical resolution data.

## Data Availability

SNPP OMPS LP version 2.6 ozone profile data are available at the NASA Goddard Earth Sciences Data and Information Services Center (GES DISC): https://disc.gsfc.nasa.gov/

MLS data are available at the NASA Goddard Earth Sciences Data and Information Services Center (GES DISC): https://disc.gsfc.nasa.gov/



ACE-FTS data are available via the ACE/SCISAT database: https://databace.scisat.ca/

SAGE III/ISS data are available at the NASA Langley Atmospheric Science Data Center (ASDC): https://asdc.larc.nasa.gov/

Ozone sonde data are available at the NASA Goddard Atmospheric composition Validation Data Center (AVDC): https://avdc.gsfc.nasa.gov/

Mauna Loa lidar data are available via the Network for the Detection of Atmospheric Composition Change (NDACC): https://ndacc.larc.nasa.gov/

OMPS NP data are available at the NASA Goddard Earth Sciences Data and Information Services Center (GES DISC): https://disc.gsfc.nasa.gov/

**Author contribution**

NK directed the work, NR and NK devised the comparison methodology, NR carried out the work and NK and NR analyzed the results. SD and YJ performed the comparisons between OMPS LP and the MLO lidar station. NR prepared the manuscript with contributions from all co-authors.

**Competing interests**

Some of the authors are members of the editorial board of Atmospheric Measurement Techniques.

**Acknowledgements**

This research is supported by the GESTAR II Cooperative Agreement with NASA Goddard Space Flight Center

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
