# Peer review of "Validation of SNPP OMPS limb profiler version 2.6 ozone profile"

_EGUsphere, 2025_

## Referee Comment (RC1)

This manuscript provides a comprehensive overview of the validation of OMPS LP v2.6 ozone profiles, discussing biases with comparative observations and the long-term stability of the SNPP dataset. This work is an extension and integration of the Kramarova et al. 2024 study, which presented the OMPS LP v2.6 retrieval. The paper is well-written and well-structured, presenting the results of the validation in a clear way. Particularly interesting is the focus provided in several parts of the paper on finding a valuable set of correlative data to be used for validation, in a future with fewer limb observations available.

I have a few minor comments to the manuscript, which are listed below and a few technical corrections.

- The methodology used to compare OMPS LP with ozonesonde is unclear. It seems that you haven't used averaging kernels to match the vertical resolution of the two profiles (as done with OMPS NP), is this right? In that case how have you performed the interpolation/smoothing of the sonde profiles? Some more details in this regard are needed.
- I would also suggest to include a couple of more references in the introduction, as only the WMO assessment (2022) is used. For example, other comprehensive studies on stratospheric trends and uncertainties are the LOTUS assessment in 2019 and Godin-Beekmann et al. 2022. A recent study involving observations and models on lower stratospheric trends is the paper by Benito-Barca et al. 2025.
- Section 2 focuses briefly on the instrument and on the retrieval, introducing the two versions v2.5 and 2.5. For this reason, I would add "and retrieval description" to the title of the section. It is also not clear from the first paragraph that you are going to use only SNPP in this paper. In the second paragraph, would it be possible to distinguish between the improvements in L1 data and the changes in the retrieval settings between v2.5 and v2.6?
- The authors provide an insight into the approaching future, with MLS and SAGE III retiring soon. At the same time, new OMPS instruments are going to be launched. In this perspective, an overview of other instruments that are going to be designed or launched in the next years would be interesting. For example, you could mention the upcoming ALTIUS mission in the introduction or in the conclusions: for this mission, OMPS will serve as a reference, making it even more important to characterize its long-term stability and biases.
- Since you mentioned in Sect. 5.1 a comparison of the correlation results with v2.5, you could also provide a short comparison between the biases found in version v2.5 w.r.t. v2.6. I am also wondering what is the difference between panel (c) of Fig. 4 to panel (a) of Kramarova 2024. Is it only the considered period? The biases appear smaller and more negative in this manuscript. To help visualize the biases, I suggest reducing the color bar extension for the first three panels of Fig. 4, e.g. to ±30%.
- Can you shortly clarify the drift computation methodology? Are you computing differences for each collocation OMPS to correlative profiles, then averaging these differences on a monthly basis, removing the seasonal cycle and finally computing the linear trend? Regarding the drift, I think it would be valuable information to include an estimation of the threshold needed to confidently detect trends in the stratosphere over the last two decades, as they are often on the order of 1-2 % per decade.
- For the comparison in polar regions, can you mention if you used a filter for polar mesospheric clouds? Have you considered any restrictions related to potential vorticity to exclude cases with collocations within/outside the polar vortex?

**Technical corrections**
L20: I would remove "the" from "the retrieval algorithm".
L30: Also here I would remove "the" from "the OMPS LP".

L41-43: Possible re-formulation of the sentence: "These increases are consistent with model simulation showing that they arise from a combination of ozone-depleting substances concentrations and decreasing upper-stratospheric temperatures, driven by increasing CO2".

L48: "and so trends have large uncertainties" → "leading to large uncertainties in trends".

L61: I would remove "when validating such data"

L85: "which is more pronounced" → "which was more pronounced".

L101: Possibly mention also that the altitude range over which ozonesondes can be used for validation is limited to about 30 km.

L112: Is the period until April 2024 or June? For lidar December 2024 is mentioned.

L130: "with which to compare with" → "to use for the comparison with"

L152: Since the v6 became recently available and you also mention it, I would avoid saying "the latest version".

L175-177: I find the two sentences in these two lines very similar: isn't the accuracy estimated by the comparison with other data sets?

L216: It is Fig. 4 not 1.

L257: The sentence is not very clear to me. Could it be that the variability of OMPS retrievals at the ozone peak is lower than for the other datasets?

L338: I think you mean between 20 and 30 km.

L379: I would add "above 20 km" at the end of the sentence.

L425: Maybe repeat the word between to make it less confusing: "and between OMPS LP and ozonesondes".

L484: Typo in OMPS NP.

L615: Remove , after "consistent".

*References*

Benito-Barca, Samuel, et al. "Recent lower stratospheric ozone trends in CCMI-2022 models: Role of natural variability and transport." *Journal of Geophysical Research: Atmospheres* 130.9 (2025): e2024JD042412.

Godin-Beekmann, Sophie, et al. "Updated trends of the stratospheric ozone vertical distribution in the 60° S–60° N latitude range based on the LOTUS regression model." *Atmospheric Chemistry and Physics Discussions* 2022 (2022): 1-28.

Petropavlovskikh, Irina, et al. "SPARC/IO3C/GAW report on Long-term Ozone Trends and Uncertainties in the Stratosphere." 26 Feb. 2019.

---

## Author Comment (AC1)

This manuscript provides a comprehensive overview of the validation of OMPS LP v2.6 ozone profiles, discussing biases with comparative observations and the long-term stability of the SNPP dataset. This work is an extension and integration of the Kramarova et al. 2024 study, which presented the OMPS LP v2.6 retrieval. The paper is well-written and well-structured, presenting the results of the validation in a clear way. Particularly interesting is the focus provided in several parts of the paper on finding a valuable set of correlative data to be used for validation, in a future with fewer limb observations available.

I have a few minor comments to the manuscript, which are listed below and a few technical corrections.

We would like to express our sincere gratitude to the referee for their thorough evaluation and valuable comments that help to improve the manuscript.

- The methodology used to compare OMPS LP with ozonesonde is unclear. It seems that you haven't used averaging kernels to match the vertical resolution of the two profiles (as done with OMPS NP), is this right? In that case how have you performed the interpolation/smoothing of the sonde profiles? Some more details in this regard are needed.

  For comparison to ozonesondes, the sonde profiles were interpolated onto the OMPS LP vertical grid using log-linear interpolation and no additional smoothing was applied. Since the OMPS LP vertical resolution is ~2km and the averaging kernels are quite narrow, almost delta functions, their application to the sonde profiles is not necessary. The figure below shows the effect of applying the OMPS LP averaging kernels to the sonde profiles on the OMPS-sonde differences over the 2012-2024 period for the 3 wide latitude bands considered in the paper. Applying the averaging kernels has little effect on the observed biases, with the largest difference (<3%) seen at the lowest altitudes in the mid-latitudes.

[Figure]

Figure RC1.1: OMPS LP vs Ozonesonde with (red) and without (black) OMPS LP averaging kernels (AK) applied

- I would also suggest to include a couple of more references in the introduction, as only the WMO assessment (2022) is used. For example, other comprehensive studies on stratospheric trends and uncertainties are the LOTUS assessment in 2019 and Godin-Beekmann et al. 2022. A recent study involving observations and models on lower stratospheric trends is the paper by Benito-Barca et al. 2025.

  Those references have been added to the introduction.

- Section 2 focuses briefly on the instrument and on the retrieval, introducing the two versions v2.5 and 2.5. For this reason, I would add "and retrieval description" to the title of the section. It is also not clear from the first paragraph that you are going to use only SNPP in this paper. In the second paragraph, would it be possible to distinguish between the improvements in L1 data and the changes in the retrieval settings between v2.5 and v2.6?

  o The section title has been updated as suggested.

  o We have added clarifying text at the end of the first paragraph to indicate that we are using SNPP in this paper.

  o Between version 2.5 and 2.6 many changes were made to both L1 and L2, the full details of the changes are given in Kramarova et al. 2024, we have re-worded this paragraph and moved the reference to Kramarova et al., earlier in the paragraph to more clearly point the reader to this paper for information on changes from version 2.5 to 2.6.

- The authors provide an insight into the approaching future, with MLS and SAGE III retiring soon. At the same time, new OMPS instruments are going to be launched. In this perspective, an overview of other instruments that are going to be designed or launched in the next years would be interesting. For example, you could mention the upcoming ALTIUS mission in the introduction or in the conclusions: for this mission, OMPS will serve as a reference, making it even more important to characterize its long-term stability and biases.

  We have added the text below to the end of the conclusions (section 9):

  With the potential upcoming "data desert" in satellite observations of atmospheric composition with high vertical resolution (Salawitch et al., 2025), the OMPS LP series of instruments will serve as a critical bridge connecting records from Aura MLS and SAGE III with future missions, like the ESA's Atmospheric Limb Tracker for Investigation of the Upcoming Stratosphere (ALTIUS). ALTIUS will be launched in 2027 and will carry a high-resolution spectral imager that measures in UV, VIS and NIR ranges. ALTIUS will acquire observations in 3 modes - limb scattering, solar and stellar occultation - to retrieve profiles of ozone, aerosol and other trace gases in the stratosphere and mesosphere.

- Since you mentioned in Sect. 5.1 a comparison of the correlation results with v2.5, you could also provide a short comparison between the biases found in version v2.5 w.r.t. v2.6. I am also wondering what is the difference between panel (c) of Fig. 4 to panel (a) of Kramarova 2024. Is it

only the considered period? The biases appear smaller and more negative in this manuscript. To help visualize the biases, I suggest reducing the color bar extension for the first three panels of Fig. 4, e.g. to ±30%.

- o We have added the following sentence summarizing the improvements to the biases relative to MLS for version 2.6 over 2.5 in section 5.1 line 247:

    - These biases represent an improvement over those observed between OMPS LP version 2.5 and MLS, with the largest reduction in biases seen below 31 km, where LP retrievals primarily rely on the visible triplet (Kramarova et al., 2024), there is also a reduction in vertical oscillations seen in version 2.5, particularly where the retrieval switches between UV and visible wavelengths (approximately 28-32 km).

- o Yes, the only difference between figure 4 panel (c) of this paper and figure 8 panel (a) of Kramarova 2024 is the time period used, in this paper we use April 2012 to April 2024 whereas Kramarova 2024 uses April 2012 to December 2021.

- o The color bar for Fig. 4 panels a-c has been updated to ±30% as suggested:

[Figure]

Figure RC1.2: Updated Fig. 4 panels a-c

- Can you shortly clarify the drift computation methodology? Are you computing differences for each collocation OMPS to correlative profiles, then averaging these differences on a monthly basis, removing the seasonal cycle and finally computing the linear trend? Regarding the drift, I think it would be valuable information to include an estimation of the threshold needed to confidently detect trends in the stratosphere over the last two decades, as they are often on the order of 1-2 % per decade.

- To calculate the drifts, we first calculate monthly zonal means for each instrument, we then remove the seasonal cycle from each dataset independently, the differences between these de-seasonalized monthly zonal means are then calculated, and finally we calculate the drift by fitting a linear trend to the de-seasonalized monthly zonal mean differences.

- We have added the text "In order to confidently detect long-term ozone trends in the stratosphere, a threshold stability requirement of 3% per decade for ozone stratospheric profiles has been set by the World Meteorological Organisation (WMO 2022)" to the drift discussion in the conclusions section

- For the comparison in polar regions, can you mention if you used a filter for polar mesospheric clouds? Have you considered any restrictions related to potential vorticity to exclude cases with collocations within/outside the polar vortex?

  - Yes, we have filtered out any profiles that contain polar mesospheric clouds. The OMPS LP version 2.6 ozone product contains a flag for PMC's which has been used to filter out those data.

  - Comparisons in polar regions were against MLS and OMPS NP for which we have very close coincidences, we therefore don't expect mapping measurements on equivalent latitude coordinates to produce substantial differences.

**Technical corrections**

L20: I would remove "the" from "the retrieval algorithm". - done

L30: Also here I would remove "the" from "the OMPS LP". - done

L41-43: possible re-formulation of the sentence: "These increases are consistent with model simulation showing that they arise from a combination of ozone-depleting substances concentrations and decreasing upper-stratospheric temperatures, driven by increasing CO2". - done

L48: "and so trends have large uncertainties" → "leading to large uncertainties in trends". – This sentence was removed

L61: I would remove "when validating such data" - done

L85: "which is more pronounced" → "which was more pronounced". - done

L101: Possibly mention also that the altitude range over which ozonesondes can be used for validation is limited to about 30 km. - done

L112: Is the period until April 2024 or June? For lidar December 2024 is mentioned. – This has been corrected to April 2024. MLO lidar data was only available up to December 2022 and so this is what was used in the lidar analysis.

L130: "with which to compare with" → "to use for the comparison with" - done

L152: Since the v6 became recently available and you also mention it, I would avoid saying "the latest version". – Changed to "last version"

L175-177: I find the two sentences in these two lines very similar: isn't the accuracy estimated by the comparison with other data sets? – We have removed the second sentence

L216: It is Fig. 4 not 1. - Corrected

L257: The sentence is not very clear to me. Could it be that the variability of OMPS retrievals at the ozone peak is lower than for the other datasets?

   The variability of OMPS ozone and that of the other datasets at the ozone peak are comparable. The sentence has been revised to be clearer and now reads:

       "The drop in correlations seen at around 25 km at all latitudes and against all correlative sources is likely because this is where ozone density peaks and it's variability is lower leading to weaker correlations."

L338: I think you mean between 20 and 30 km. – This has been corrected to 12 and 20 km

L379: I would add "above 20 km" at the end of the sentence. - done

L425: Maybe repeat the word between to make it less confusing: "and between OMPS LP and ozonesondes". - done

L484: Typo in OMPS NP. - Corrected

L615: Remove , after "consistent". - Done

*References*

Benito-Barca, Samuel, et al. "Recent lower stratospheric ozone trends in CCMI-2022 models: Role of natural variability and transport."*Journal of Geophysical Research: Atmospheres* 130.9 (2025): e2024JD042412.

Godin-Beekmann, Sophie, et al. "Updated trends of the stratospheric ozone vertical distribution in the 60° S–60° N latitude range based on the LOTUS regression model." *Atmospheric Chemistry and Physics Discussions* 2022 (2022): 1-28.

Petropavlovskikh, Irina, et al. "SPARC/IO3C/GAW report on Long-term Ozone Trends and Uncertainties in the Stratosphere." 26 Feb. 2019.

---

## Author Comment (AC2)

The manuscript provides a validation of the SNPP OMPS-LP v2.6 ozone dataset. Validation is done primarily with comparisons to MLS/ACE/SAGE III/ISS, but also other data sources like ozonesondes, lidar, and OMPS-NP. It is an important paper to be in the literature since OMPS-LP is poised to become the backbone of vertically resolved global ozone measurements in the near future. Overall the flow is easy to follow, presents useful information, and should be published subject to some revisions. My main suggestion is that the information could be summarized better for data users, specifically the estimated drift levels for the dataset.

We would like to express our sincere gratitude to the referee for their thorough evaluation and valuable comments that help to improve the manuscript.

**General Comments**

The choice to not degrade SAGE III/ISS to a similar resolution I don't think has a justification other than convenience. While it doesn't change any of the main conclusions of the paper in my opinion, various statements in the discussion are likely because of this choice. For example, the significantly degraded correlation with SAGE at high altitudes, or some of the increased oscillations noted in the drift analysis. I would suggest the SAGE III/ISS data be smoothed to approximately the same resolution as OMPS-LP and the analysis repeated.

In our analysis the SAGE III/ISS measurements were interpolated onto a 1 km grid for comparison with OMPS. We tested smoothing the SAGE III/ISS data by applying a 2 km boxcar average smoothing function to the profiles and the recalculated the difference profiles and drifts using the smoothed data. As can be seen from the plots below this has little impact on the observed biases between OMPS LP and SAGE III/ISS and the oscillations in the drifts are still present and are a result of the shorter time period available for SAGE III/ISS. Since these changes are minor and do not change any of the discussions or conclusions in the paper we have elected to retain the original 1 km analysis in the paper.

[Figure]

Figure RC2.1: OMPS LP vs SAGE III/ISS differences with (red) and without (black) 2 km boxcar average smoothing of SAGE III/ISS profiles for the period 2017-2024.

[Figure]

Figure RC2.2: OMPS LP vs SAGE III/ISS drifts with 2 km boxcar average smoothing applied to SAGE III/ISS profiles.

The paper is long, I don't believe it is too long, but there is a lot of information. What might be useful is to provide a summary table (something akin to the MLS data quality document tables) of observed drifts/estimated accuracy for the v2.6 data products based on the analysis done here to better provide easy to digest recommendations to users of the data.

The following summary table has been added in the conclusions section (section 9).

| Altitude (km) | 60ºS to 30ºS | | 30ºS to 30ºN | | 30ºN to 60ºN | |
|---|---|---|---|---|---|---|
| | Bias (%) | Drift (%/decade) | Bias (%) | Drift (%/decade) | Bias (%) | Drift (%/decade) |
| **15.5** | -5.84 (±2.84) | -4.74 (±0.89) | - | - | 3.34 (±0.28) | -3.50 (±0.62) |
| **20.5** | 4.32 (±1.08) | 1.87 (±3.02) | 2.32 (±3.88) | 1.14 (±2.57) | 4.14 (±0.07) | 3.44 (±1.07) |
| **25.5** | -2.31 (±1.81) | 0.44 (±1.48) | -0.86 (±0.94) | -0.40 (±2.15) | -8.15 (±1.66) | 0.93 (±0.17) |
| **30.5** | 0.65 (±9.54) | -2.38 (±0.47) | -3.90 (±6.14) | -1.41 (±0.08) | -5.29 (±6.24) | -0.54 (±0.69) |
| **35.5** | 2.61 (±0.50) | -0.22 (±0.59) | 2.58 (±0.38) | 0.91 (±0.19) | -0.43 (±0.58) | 1.24 (±1.55) |
| **40.5** | -0.21 (±1.29) | -1.75 (±1.24) | -0.35 (±1.29) | -0.22 (±0.55) | -2.02 (±1.79) | -1.06 (±1.37) |
| **45.5** | -4.57 (±0.33) | -2.12 (±3.30) | -4.07 (±0.11) | 0.22 (±1.50) | -6.76 (±0.32) | 0.54 (±2.24) |
| **50.5** | -6.48 (±0.78) | -2.11 (±1.44) | -6.61 (±0.71) | -1.16 (±0.25) | -8.41 (±0.86) | 0.11 (±2.84) |
| **55.5** | -10.59 (±25.79) | -1.21 (±2.26) | -9.43 (±15.23) | -2.32 (±0.12) | -13.18 (±21.80) | 1.59 (±4.49) |

I find the stability discussion not to be wrong, but maybe is too optimistic in some places. Specifically the statement "We therefore conclude that there is no significant systematic drift in OMPS LP version 2.6 ozone for the period 2012 to 2024 and that it is suitable for use in ozone trend studies." which may not technically be incorrect, but the numbers presented are 2%/decade from 20-50 km and becoming larger outside. Statements like "We find small drifts ... of less than 2%/decade ..." may not that be useful to potential users of the data trying to attribute changes of 1%/decade, which is the level that most trend studies would be looking at. I think this is solved if a summary table is included with estimated drift levels so the user can decide if it is useful for their application, as well as maybe a softening of the blanket statement that the dataset is suitable for ozone trend studies.

A summary table has been added as mentioned above. We have also added an estimation of the threshold needed to confidently detect trends in the stratosphere as suggested by referee 1 and changed the statement mentioned above to read "We therefore conclude that there is no significant systematic drift in OMPS LP version 2.6 ozone for the period 2012 to 2024 and that OMPS LP data meets current WMO requirements for long-term stratospheric ozone trend studies." We have also removed the subjective word "small" from the discussion.

**Specific Comments**

l. 98. There should be a more detailed explanation of how this study differs from the v2.6 validation already done by Kramarova et al. (2024)

We have added the following sentence to section 3:

"This study builds on Kramarova et al. (2024) which compared OMPS LP version 2.6 to MLS for the period 2012-2021 to include other sources of correlative data and extend the evaluation period to April 2024."

l. 102. The motivation/wording here seems like a repeat of what is at the start of Sec. 6, and the Sec. 6 wording does a better job of motivating how OMPS-LP could be validated going forward. On the same topic, I know it is outside the scope of the paper, but there is no mention anywhere (that I could find) of the potential of validating/cross-calibrating OMPS-LP based on the overlap between NPP and N21 or future satellites.

Although the wording is indeed similar, in this section we are advocating for the use of solar occultation instruments whilst they are available and ozonesondes, whereas in section 6 we are looking forward to a time when there are no other high vertical resolution satellite observations available for which to validate against.

It is true that there should be some overlap between successive OMPS LP instruments and we will use this to cross-calibrate/validate them. We have added the following sentence to the conclusions section:

"There will also be some overlap between successive OMPS LP instruments which we can exploit in order to cross-calibrate/validate them, this will enable us to determine any bias offsets between them."

l. 112. Here the evaluation period is stated as ending June 2024, but earlier it was April 2024?

This has been corrected to say April 2024

l. 130. At this point the co-location criteria has not been stated, but there are figures showing coincidences which is slightly confusing.

We have added "We only use observations co-located with OMPS measurements (see Section 4)" to this sentence to direct the reader to the co-location criteria.

l. 150. I believe SAGE III/ISS does extend to +/- 70 degrees latitude? albeit for a very limited time of the year

This has been changed to say "approximately 70 degrees"

l. 180. I know in previous studies since Aura and NPP are in similar orbits you can get almost perfect coincidences between MLS and OMPS-LP. Is that is what is done here? Or is the same criteria applied to MLS?

The same coincidence criteria is applied to MLS, however, we always take the closest matching profile within that criteria, so for MLS there are times where there are almost perfect coincidences and these are used.

l. 184. "The only time criterion is that the profiles be on the same day..." is this actually what is done or is it a +/- 12 hour window around the observation time?

In this study we take comparison profiles from that same day, and not from a +/- 12 hour window.

l. 186. "... do not account for the small differences in the vertical resolution..." this is true for MLS and ACE, but the SAGE vertical resolution is significantly better than OMPS-LP

As described above, we tested smoothing the SAGE III/ISS profiles but this had little effect on the biases and drifts and our overall conclusions.

l. 203. "To the 1:30 pm local solar time ..." the measurement time is only 1:30 pm at the equator, presumably you mean to the actual measurement time of OMPS-LP?

That is correct, the reference to 1:30 pm has been removed and the sentence now reads "to adjust both ACE-FTS and SAGE III/ISS observations to the measurement time of OMPS LP"

l. 206. When I read this paragraph I thought it meant that we would only be seeing results from the wide latitude bands, but that is not the case. I would reword it or move it after the 5 degree zonal means comparisons are done

The paragraph has been reworded as below:

"Initially matched profiles were averaged into 5 degree zonal means for comparison. In addition, owing to limited data coverage from correlative solar occultation satellite observations (see Figs. 2 & 3), the data were further averaged into 3 wide latitude bands to increase the number of

observations in each bin for comparison statistics. These bands are 30⁰S-60⁰S, 30⁰S-30⁰N and 30⁰N-60⁰N and exclude the polar regions. "

Sec 5.1: I find the systematic difference between MLS and SAGE interesting here, specifically in Figure 4 you could interpret it as MLS being ~5% higher than SAGE almost everywhere.  This is quite different than say what is reported in the Wang et al. validation of SAGE.  I know it's different data versions etc, and doesn't have anything to do with OMPS-LP, but I'm wondering if you noticed this and is it expected.

The differences seen in Figure 4 indicate that SAGE is higher than MLS by ~5%, since here we are plotting OMPS-MLS and OMPS-SAGE. If we do the same analysis with SAGE version 5.3 (see figure below) then the comparison is closer to MLS, and is consistent with Wang et al. (2020). However, when we switch to SAGE version 6, the SAGE/OMPS/MLS difference increases, and this is consistent with what is stated in the data users guide for SAGE III/ISS version 6 which says that ozone is increased by ~2% in this version.

[Figure]

Figure RC2.3: Differences between OMPS LP and correlative measurements using SAGE III/ISS version 5.3 data (red) instead of version 6.

Fig. 4: I find it hard to quantify the differences from the color plots since the scale is so large.  Perhaps some black contour lines could be added to the for example +/- 10% difference levels to guide the reader?

The color scale for Figure 4 has been updated to ±30% as suggested by referee 1:

[Figure]

Figure RC2.4: Updated Fig. 4 panels a-c

l. 237. Does the altitude biases correlate with the changing wavelengths used in the retrieval in altitude?

In version 2.6, the number of UV pairs has been increased to 6, compared with 3 UV pairs used in v2.5. The vertical range where the algorithm suppresses the contributions from shorter wavelengths is now dynamical in v2.6, whereas it was static in v2.5. As a result, we do not see clear changes in bias structures that can be directly attributed to a wavelength switch, except in the tropical middle stratosphere ~ 30-32 km, where the algorithm transitions from the longest UV pair (322/356 nm) to the VIS triplet (510/606/675 nm). However, in this part of the paper we describe vertical pattern of biases in the middle and lower stratosphere, where retrievals solely relay on the only VIS triplet.

l. 250. "degraded precision and increased noise for SAGE III/ISS measurements...": yes, but a lot of this is probably because of using the raw 0.5 km SAGE measurements instead of degrading it down to the resolution of the other measurements

In our analysis the SAGE III/ISS measurements were interpolated onto a 1 km grid for comparison with OMPS, Wang et al. also noted increased noise and biases in SAGE III/ISS ozone at higher altitudes, however, we agree that there may be a contribution from the differences in vertical resolution, and we have now added the statement "it should be noted that, although we interpolate SAGE III/ISS observations from a 0.5 km to a 1 km vertical grid, we have not degraded the SAGE III/ISS profiles down to the resolution of OMPS LP, and this may also contribute to the lower correlations at higher altitudes" after this line.

Fig. 5: Is there some motivation for the line at 0.75?

This was to be consistent with Figure 11 in Kramarova et al. (2018) which showed correlations for OMPS LP version 2.5, to allow the reader to more easily compare the two figures

l. 307. "However, these biases are not seen when compared to MLS." is a possible explanation uncertainties in the diurnal scaling?

The biases are present both before and after diurnal scaling (see figure below) and so we don't believe that these biases are related to the diurnal correction.

[Figure]

Figure RC2.5: Seasonal cycle biases between OMPS LP and SAGE III/ISS with (top) and without (bottom) diurnal correction (DI).

Figure 8: The scales here are really quite large, here we are extending to +/- 20%/decade, when observable ozone trends are ~1%/decade

The scale in Fig. 8 was optimized to illustrate drifts and associated error bars over the large vertical range. The figure has been updated (see below) to add vertical dashed-dotted lines indicating 0.3%year drifts, which is the WMO stability threshold value for stratospheric ozone trend studies.

[Figure]

Figure RC2.6: Updated Figure 8 with 0.3%/year lines added.

l. 497. Here the drift is calculated until the end of 2021 at low altitudes, presumably because of the Hunga influence. But this wasn't done in comparisons with the other instruments, so it is an issue with OMPS NP? It seems odd.

The same was done for the other instruments. Panels d-f in Fig. 8 show the drifts calculated to the end of 2021 at low altitudes. This was indeed done because of the Hunga influence and is described in the text in the paragraph beginning at line 348 in the original manuscript.

l. 522. "Previously, we limited our comparisons geographically to exclude polar regions" I think some of the comparisons with MLS previously did extend to 80?

Whilst it is true that some MLS/OMPS differences extended to 80 degrees for the 5 degree zonal mean plot in Fig. 4 panel c, all of the analysis was limited to 60S to 60N.

Figure 12.  It is odd here to use two significantly different scales, 10% for the NP comparisons and 50% for the MLS comparisons.  Also error bars are only shown for the NP comparisons?

> The plot has been updated to rescale the MLS comparisons to be consistent with the NP comparisons (see below). The error bars are there for MLS but they are very small.

[Figure]

Figure RC2.7: Updated Figure 12.

Figure 13. Here we are also back to using the full time period with NP drift calculations instead of just to 2021

> We used the full time period here as there was very little or no effect of the Hunga eruption in the polar regions for comparisons with MLS and NP.

Sec 8. This section is quite short, and all of the information is in the other sections in some shape or form. I would ask the authors to consider if it could be removed.

> The section was removed and incorporated into the conclusions section.

**Technical Corrections**

l. 484. "OMP SNP" -> "OMPS NP" - Corrected